# Structural consistency in AI governance: A PMC index assessment with evidence from China's central-level policies

Simeng Zhang, Tao Zhang, Xi Wang *

Faculty of Humanities and Social Sciences, Macao Polytechnic University, Macao, People's Republic of China

* xwang@mpu.edu.mo

## Abstract

The structural coherence of policy design has become an increasingly important issue in artificial intelligence (AI) governance. This study evaluates the structural consistency of China's central-level AI policies issued between 2016 and 2025 (n = 54). It combines text mining to identify high-frequency policy terms and semantic co-occurrence patterns with a Policy Modeling Consistency (PMC) index framework comprising nine primary and forty-three secondary indicators. Five representative policies are then selected for detailed quantitative evaluation and visual comparison. The results show that China's AI policy system is generally well structured, but still exhibits notable weaknesses in temporal planning, intergovernmental coordination, and incentive design. In particular, long-term policy supply remains limited, vertical coordination mechanisms are insufficiently institutionalized, and policy instruments are unevenly configured across key support dimensions. These findings suggest that future policy improvement should focus on strengthening medium- and long-term planning, enhancing coordination across governance levels, and improving the integrated design of policy instruments. Methodologically, the study demonstrates a reproducible analytical framework linking text analysis, indicator construction, quantitative evaluation, and visualization. It contributes to the literature by moving from thematic description toward structural assessment in the study of AI governance.

## Introduction

Since the release of the "Internet Plus" Artificial Intelligence Three-Year Action Implementation Plan [1] and the New Generation Artificial Intelligence Development Plan [2], the Chinese government has continuously strengthened policy supply in the field of artificial intelligence (AI). Over time, it has gradually developed a systematic policy framework encompassing core technology research and development, infrastructure construction, talent cultivation, ethical governance, and industrial ecosystem improvement. Entering the period of the 14th Five-Year Plan, AI has become not only a vital

**Data availability statement:** All relevant data are within the paper and its Supporting information files.

**Funding:** The author(s) received no specific funding for this work.

**Competing interests:** The authors have declared that no competing interests exist.

component of China's strategic emerging industries but also an embedded element in national initiatives such as Digital China, Intelligent Manufacturing, and East-to-West Computing Resource Transfer. These developments have positioned AI as a critical pillar of China's technological strategy and governance modernization. In 2025, China's Government Work Report formally proposed the "AI+" strategy for the first time, underscoring AI's central role in advancing new productive forces and reaffirming its strategic priority in China's path toward modernization [3].

As the global wave of technological revolution and industrial transformation accelerates, AI has evolved from an auxiliary technology into a general-purpose technology with far-reaching implications for economic development, public administration, and social organization. It increasingly reshapes production modes, governance mechanisms, and institutional arrangements. Under this trend, China's AI industry has entered a critical stage in which policy guidance and industrial transformation advance in parallel. Since the 13th Five-Year Plan for National Scientific and Technological Innovation [4] elevated AI to the level of national strategic priority, related policies have continued to expand under the dual logics of "new infrastructure" and digital transformation. As a result, China's AI policy system now exhibits extensive content coverage, complex instrument combinations, and diversified implementation arrangements, reflecting both the forward-looking intent of the state and the adaptive evolution of policy tools under multiple governance objectives.

At the same time, the intrinsic uncertainty, cross-domain integration, and ethical risks associated with AI technologies have generated increasing tension between policy responsiveness and institutional coordination. In this context, AI policymaking should not be understood as a simple accumulation of fragmented policy texts or symbolic declarations. Rather, it should be viewed as part of a broader governance process that requires coherence among policy goals, instruments, implementation pathways, and regulatory arrangements. In this study, artificial intelligence governance is understood as the set of institutional, strategic, and regulatory arrangements through which the state guides, supports, coordinates, and constrains the development and application of AI. In the context of China's central-level policy system, AI governance is reflected not only in the promotion of technological innovation and industrial upgrading, but also in the design of regulatory frameworks, implementation mechanisms, incentive instruments, and multi-level coordination arrangements. From this perspective, AI governance is not merely a matter of issuing individual policy documents; rather, it concerns whether key policy elements are configured in a coherent and internally consistent way across major governance dimensions.

Existing scholarship on AI policy has primarily focused on content mining, thematic evolution, policy instrument typologies, or the effectiveness of individual policy measures. Although these studies have generated valuable insights into policy themes and functional patterns, relatively limited attention has been paid to the structural consistency and internal coordination of AI policy systems as a whole. In particular, at the macro-policy level, few studies have employed quantitative or structural models to examine whether policy architectures remain coherent across dimensions such as policy nature, timeliness, content, support mechanisms, target actors, and

governance scope. This gap is especially significant in the case of China, where central-level AI policy documents embody not only strategic priorities but also the governance logic through which the state organizes technological development, regulatory design, and institutional implementation.

Against this background, evaluating the structural consistency of central-level AI policies provides an analytically meaningful entry point for understanding the institutional logic of China's AI governance framework. Rather than treating policy documents as isolated texts, such an approach makes it possible to examine how governance intentions are translated into structured policy configurations. In this regard, the Policy Modeling Consistency (PMC) Index Model offers a useful framework for assessing the degree of coordination among multidimensional policy elements. Based on a corpus of central-level AI policy documents issued between 2016 and 2025, this study develops a PMC-based structural evaluation framework to examine the consistency of China's AI policy design. By combining policy text analysis with structured policy evaluation, the study aims to provide a more systematic and comparable approach to understanding the structural features of AI governance in China, while also contributing a transparent analytical framework for future comparative policy research.

## Literature review

### Research on artificial intelligence governance and policy

Research on artificial intelligence (AI) policy has gradually shifted from a techno-economic perspective toward a broader governance-oriented perspective. Early studies mainly treated AI policy as an institutional instrument for mobilizing innovation resources, improving industrial competitiveness, and supporting national strategic development. As AI technologies have become increasingly embedded in economic and social systems, however, scholarly attention has expanded to questions of ethics, regulation, public governance, and social legitimacy. In this context, AI is no longer understood merely as a technological agenda, but as a governance issue involving strategic planning, institutional design, and social coordination [5–11].

At the national strategic level, existing studies show that AI policy is closely connected to the socio-technical construction of state priorities and political legitimacy. Comparative analyses suggest that AI strategies vary substantially across political and developmental contexts: developed economies tend to emphasize ethical governance, transparency, explainability, and human-centric values, whereas emerging economies more often prioritize technological upgrading, industrial transformation, and modernization goals [7,8]. At the same time, many studies point out that current AI policy frameworks still suffer from major weaknesses, including ambiguous concepts, insufficient implementation detail, and limited evaluation mechanisms [6,8,9]. These limitations indicate that AI governance cannot be adequately understood through policy declarations alone, but must also be examined in terms of whether policy systems are institutionally coherent and operationally actionable.

At the level of ethics and social governance, scholars have further highlighted the tension between regulatory discourse and technological reality. Studies show that AI experts often adopt function-oriented understandings of AI, whereas policymakers frequently frame AI in anthropomorphic terms, thereby shifting policy attention toward future imaginaries rather than currently deployed systems [9]. Related research in emerging and developed contexts alike has shown that AI governance frameworks continue to face structural challenges in addressing accountability, transparency, risk management, and cross-sector coordination [10,11]. In addition, studies on global AI indices and benchmarking practices suggest that national AI trajectories may become locked into pre-existing competitiveness frameworks, thereby narrowing the space for alternative governance models [12].

With the increasing use of computational methods, AI policy research has also moved from qualitative discourse analysis toward more data-oriented approaches. Studies using machine learning, natural language processing, and semantic analysis have identified latent themes, policy priorities, and discursive patterns in AI policy texts [13–15]. These studies have significantly enriched understanding of the content and orientation of AI policy systems. However, the existing

 

literature remains largely content-oriented. Most studies focus on policy themes, ethical concerns, discourse patterns, or policy tool classifications, while relatively limited attention has been paid to the structural consistency and internal coordination of AI policy systems. This limitation is particularly important in the case of China, where central-level AI policy documents function not only as strategic texts but also as key institutional carriers of AI governance.

### Research on the PMC index model for policy evaluation

The Policy Modeling Consistency (PMC) Index Model emerged within the broader evolution of policy evaluation from result-oriented assessment toward more systematic, reflexive, and integrative forms of analysis [16,17]. Rooted in the Omnia Mobilis hypothesis, the PMC model evaluates the internal consistency of policy texts by constructing a multi-input–multi-output matrix that captures the degree of coordination among policy objectives, instruments, and implementation elements [18]. Compared with purely descriptive approaches, the PMC framework offers a structured, parameterized, and visualized means of examining the internal logical configuration of policy design.

Existing studies have applied the PMC model to a wide range of policy fields, including ecological governance, industrial development, regional policy, public service reform, and social governance [19–24]. Across these applications, the model has demonstrated particular value in identifying structural imbalances within policy design, such as weaknesses in timeliness, incentive mechanisms, implementation support, or cross-sector coordination. In this sense, the PMC model is especially useful when the analytical objective is not only to identify policy content, but also to evaluate whether policy structures are internally coherent and logically coordinated.

Despite its growing use in policy evaluation, the application of the PMC framework to AI policy remains limited. This is an important gap, because AI governance is inherently multidimensional, involving innovation support, industrial development, regulatory design, ethical governance, implementation arrangements, and coordination across governance levels. These characteristics make AI policy especially suitable for PMC-based structural evaluation.

### Summary

Overall, existing studies have generated important insights into AI governance, policy discourse, ethical regulation, and policy instrument design, but they have rarely examined the structural consistency of AI policy systems in a systematic way. At the same time, the PMC Index Model has shown clear methodological value in evaluating the internal coordination of policy structures, yet its application to AI policy remains underdeveloped.

Accordingly, this study introduces the PMC model into the analysis of China's central-level AI policies in order to move from thematic description toward structural evaluation. By examining policy consistency across multiple dimensions, the study seeks to reveal the internal logic, coordination patterns, and potential structural imbalances of China's AI governance framework.

## Design and construction of the PMC index model for quantitative evaluation of China's AI policies

### Research framework

The overall logical foundation of this study lies in the fact that artificial intelligence (AI) policy has increasingly become a core component of China's national strategy and governance practice. How to conduct a systematic and quantitative evaluation of the policy texts issued at the central government level constitutes not only a methodological challenge within academic research but also a practical necessity for policy refinement. Anchored in this awareness, the study first reviews existing research on AI policy and policy evaluation methods, identifying that current analytical approaches remain limited and have yet to fully reveal the structural characteristics and developmental logic of China's AI policy system.

Accordingly, this study establishes the Policy Modeling Consistency (PMC) Index Model as its core methodological framework, supplemented by text-mining techniques for auxiliary content analysis. A total of fifty-four AI-related policy

documents issued by China's central government between 2016 and 2025 were selected as the sample, and the ROS-TCM6.0 software was employed to perform word frequency and semantic analysis. On this basis, a PMC index variable indicator system was constructed to ensure the rationality and reliability of the indicators, thereby minimizing subjectivity in variable formulation and supporting the objective identification of policy content within China's AI policy corpus.

Further, the PMC index model is applied to conduct a quantitative evaluation of the sampled policies, and the results are visualized through three-dimensional surface mapping to illustrate the degree of coordination and divergence across different dimensions of China's AI policy design. Based on the evaluation outcomes, the study proceeds to analyze and discuss the findings, identifying both the strengths and deficiencies of existing AI policy arrangements and offering targeted recommendations for future policy optimization. The entire research process unfolds in a progressive and reproducible manner, encompassing the origin of the study, literature review, methodological determination, data selection, model construction, result visualization, and interpretive discussion, thereby forming a coherent and verifiable research pathway. Figure 1 presents the overall research framework.

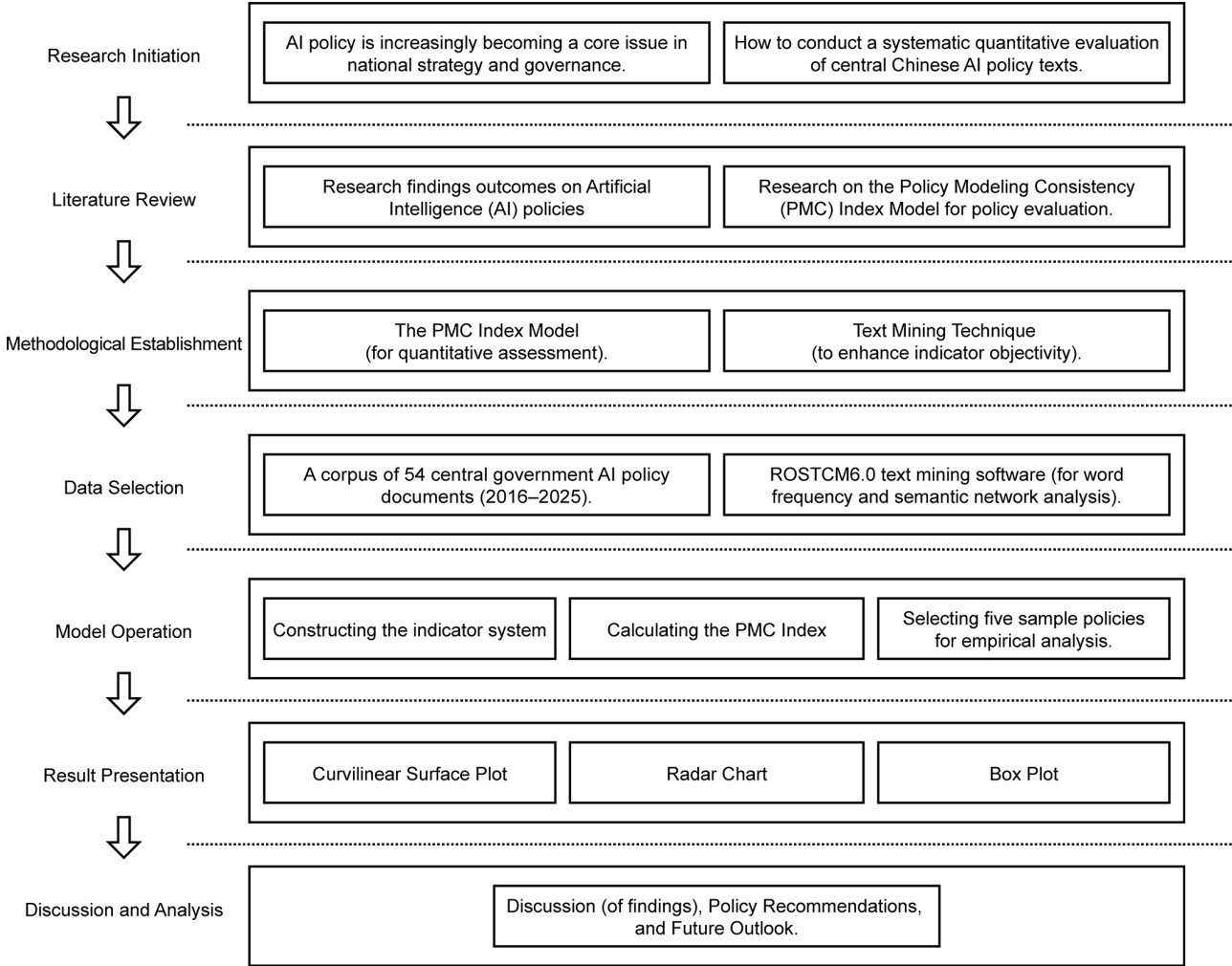

**Fig 1. Research framework of the PMC index model for the quantitative evaluation of China's artificial intelligence policies.**

## Research method

This study combines text mining with the Policy Modeling Consistency (PMC) Index Model to evaluate the structural consistency of China's central-level artificial intelligence (AI) policies. The methodological logic is sequential rather than purely automated. Text mining is first used to explore the thematic features of the policy corpus and to provide an empirical basis for variable construction. On this basis, the PMC indicator system is developed through the integration of prior policy-evaluation research, corpus-based interpretation, and researcher judgment. The model is then applied to representative policy cases through structured binary coding, PMC index calculation, and visual analysis.

The PMC Index Model serves as the core analytical framework of the study. Proposed by M. A. R. Estrada (2011), the model was designed to evaluate the internal consistency of policy texts through a structured multi-input–multi-output framework [25]. Its analytical value lies in its ability to transform complex policy content into a measurable configuration of variables, thereby revealing the strengths, weaknesses, and internal coordination of policy design. In the present study, the PMC model is used not simply to rank policies, but to assess whether key elements of AI governance—such as strategic orientation, policy content, incentive arrangements, target actors, and levels of influence—are configured in a coherent and balanced manner.

More specifically, the research process consists of four stages. First, a corpus of 54 AI-related policy documents issued by China's central government between 2016 and 2025 was constructed through systematic retrieval, screening, and validation. Second, the full texts of these policy documents were merged into a single corpus and processed through Chinese word segmentation, word-frequency analysis, and semantic co-occurrence analysis. This stage was conducted primarily to identify recurrent policy terms, thematic clusters, and semantic associations within the corpus. Third, based on the text-mining results, relevant policy-evaluation literature, and close reading of the policy texts, a PMC indicator system containing nine primary variables and forty-three secondary variables was established. Fourth, five representative policies were selected for detailed PMC evaluation. Each secondary variable was coded using binary assignment criteria (1 = Yes; 0 = No) according to the predefined scoring matrix, and the resulting values were used to calculate the PMC Index and generate visual outputs.

Accordingly, the text-mining component of the study should be understood as supportive rather than determinative. It was used to reveal the semantic structure of the policy corpus and to assist in the identification of analytically relevant dimensions, but it did not automatically generate the final policy scores. The final construction of the indicator system, the interpretation of policy elements, and the assignment of binary values all relied on structured researcher judgment guided by explicit coding rules. In this sense, the study combines automated corpus processing with interpretive policy analysis in order to achieve both analytical transparency and methodological rigor.

As illustrated in Fig 2, the analytical procedure can be summarized into four main steps. First, the evaluation indicators and corresponding coding criteria were established, including the definition of primary and secondary variables and the

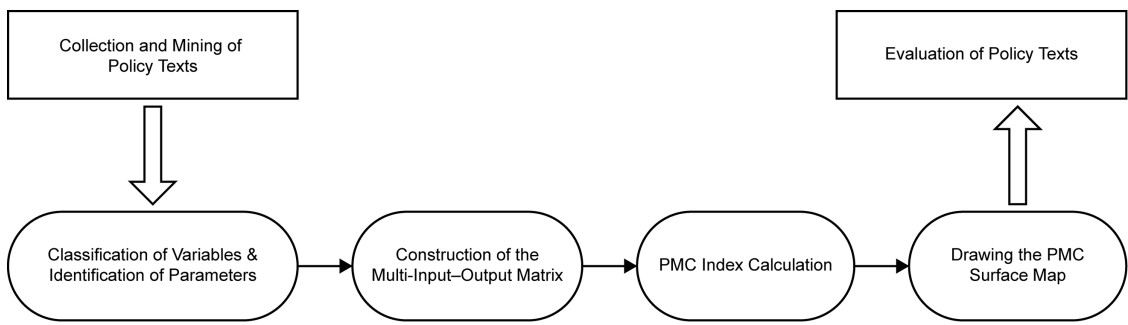

**Fig 2. Construction process of the PMC index model for China's artificial intelligence policies.**

scoring basis for each secondary indicator. Second, the policy corpus was processed through text mining in order to identify high-frequency terms, semantic associations, and major thematic tendencies relevant to AI governance. Third, based on these results and the predefined scoring matrix, the selected representative policies were organized into a multi-input–output structure and coded using binary assignment criteria. Fourth, the PMC Index was calculated for each policy, and the results were visualized through PMC surface mapping to reveal the relative strengths, weaknesses, and structural characteristics of policy design.

Taken together, these steps form a transparent and reproducible analytical workflow that combines automated corpus processing with structured policy interpretation, thereby enabling the systematic evaluation of the structural consistency of China's AI policy system.The reporting of the text-mining procedure was further refined with reference to prior methodological studies on transparent text-analytic workflows, including Danesh et al. (2021) [26].

## Data sources

This study draws on policy documents related to artificial intelligence (AI) issued by China's central government between 2016 and 2025 (up to August 31, 2025), comprising a total of 54 valid samples. The choice of 2016 as the starting point is based on two main considerations. First, although no overarching AI policy framework had yet been released that year, several national development and sectoral plans—such as the 13th Five-Year Plan for National Informatization, the 13th Five-Year Plan for Strategic Emerging Industries Development, and the "Internet Plus" Artificial Intelligence Three-Year Action Implementation Plan—explicitly referenced AI development goals or directions in certain sections, signifying the initial inclusion of AI within China's national policy system. Second, the formal issuance of the New Generation Artificial Intelligence Development Plan by the State Council in 2017 marked the beginning of a new phase characterized by systematic planning and strategic deployment of AI policy in China. Therefore, setting 2016 as the baseline year carries both retrospective and practical significance, providing a comprehensive foundation for analyzing the trajectory of policy evolution.

The study covers AI policy documents released between 2016 and 2025, with the specific data retrieval period spanning August 30–31, 2025. Policy retrieval and screening were conducted primarily through authoritative official sources, including the Chinese Government Website (www.gov.cn) and PKUlaw (pkulaw.com). During the cross-checking and validation process of AI-related policies, the following selection principles were applied:

1. Keyword relevance: Policies were identified using "Artificial Intelligence" and "New Generation Information Technology" as core keywords. Only texts exhibiting strategic guidance, institutional construction, or industrial support functions were retained, ensuring strong thematic relevance to the research topic.

2. Authority and validity of data sources: All policy texts were obtained from official and authoritative central government or ministerial websites, ensuring legality, reliability, and completeness. Only currently valid policies were included, while expired or superseded documents were excluded.

3. Normative document types: The study primarily selected formal and regulatory policy documents, including plans, opinions, guidelines, notices, implementation schemes, and development strategies. In contrast, non-normative materials—such as leaders' speeches, internal memos, training notices, meeting summaries, annual reports, or public name lists—were excluded to maintain analytical rigor and policy comparability.

Through this systematic process of retrieval, screening, and validation, the final corpus was constructed to reflect the central policy architecture of China's AI governance framework. For purposes of transparency and reproducibility, the complete list of the 54 policy documents is provided in S1 File in S1 Data.

According to the selection principles described above, a total of 54 valid policy documents were included in the sample. As shown in Table 1 and Fig 3, the annual distribution of China's AI policies exhibits distinct stage characteristics. The release of the New Generation Artificial Intelligence Development Plan in 2017 marked a critical turning point, after

**Table 1. Selected sample of China's artificial intelligence policy documents.**

| No. | Title of Policy Document | Year of Issue | Issuing Authority |
|---|---|---|---|
| 1 | Notice on the Issuance of the 13th Five-Year Plan for the Development of Strategic Emerging Industries | 2016 | State Council |
| 2 | Guiding Opinions on Accelerating Scenario-Based Innovation to Promote High-Level Application of Artificial Intelligence and High-Quality Economic Development | 2022 | Ministry of Science and Technology, Ministry of Education, Ministry of Industry and Information Technology, Ministry of Transport, Ministry of Agriculture and Rural Affairs, National Health Commission |
| 3 | Notice on Supporting the Construction of New Generation Artificial Intelligence Demonstration Application Scenarios | 2022 | Ministry of Science and Technology of China |
| 4 | Global Artificial Intelligence Governance Initiative | 2023 | Cyberspace Administration of China |
| 5 | Opinions on Deeply Implementing the "AI+" Action Plan | 2025 | State Council |

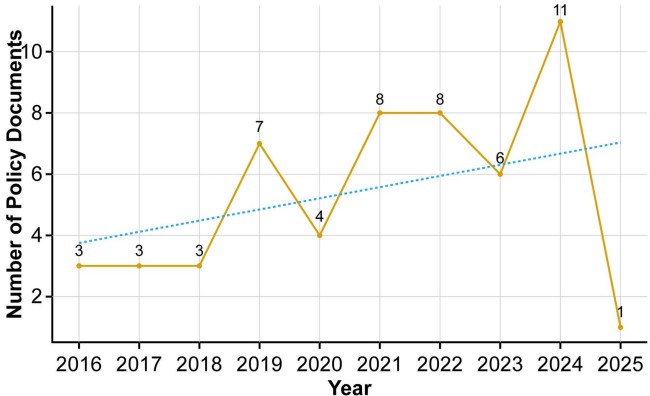

**Fig 3. Trend of the number of artificial intelligence policy documents issued by the Chinese central government (2016–2025).** Note: The dashed line indicates the overall trend in annual policy issuance.

which the number of AI-related policy issuances increased significantly. Two noticeable peaks occurred in 2018 and 2021, reflecting a cyclical policy trajectory characterized by planning, implementation, and upgrading.

The publication of the Opinions on Deeply Implementing the "AI+" Action Plan by the State Council in 2025 represents the most recent central-level document, signifying China's strategic deployment of AI integration within the broader context of "new productive forces." This policy continues the institutional trajectory established by previous documents and implies the beginning of a new phase of systemic policy upgrading.

Overall, the time-series characteristics of the sample demonstrate both continuity and representativeness, providing a comprehensive reflection of the evolutionary stages and structural features of China's AI policy system. These publicly available policy documents constitute the sole data source of the study and provide a solid empirical foundation for the subsequent phases of text mining and structural evaluation within the PMC index modeling framework. Accordingly, all coding, interpretation, and scoring procedures were conducted by the authors within a document-analysis framework. No human participants, external experts, interviewees, survey respondents, personal data, medical records, biological materials, or field research were involved at any stage of the research; therefore, ethics approval and informed consent were not required.

## Variable setting and parameter identification for the valuation of China's AI policies

**Policy text mining and word frequency analysis.** The preprocessing of policy texts constituted the starting point for variable construction in this study. The input for text mining consisted of the full texts of the 54 central-level AI policy documents included in the corpus, rather than only titles or selected excerpts. Text-mining techniques were employed to identify recurrent terms, semantic associations, and major thematic tendencies within the corpus, thereby providing an empirical basis for subsequent variable setting. In this sense, text mining was used to support corpus exploration and variable identification, rather than to automatically generate the final PMC scores.

Specifically, the text-mining procedures were conducted using ROSTCM6.0 software. First, all 54 AI-related policy documents were merged into a single plain-text corpus and imported into the software for Chinese word segmentation. In the word-segmentation stage, this study used the lexical resource available in the ROSTCM6.0 environment as the initial segmentation basis, and further refined the segmentation output through corpus-oriented manual checking, selective retention of AI-relevant policy expressions, and filtering. Based on the initial segmentation results, structurally repetitive or analytically uninformative expressions—such as article numbers, chapter labels, and procedural terms—were identified and removed, while policy-relevant terms associated with AI development and governance were retained and better aligned with the analytical purpose of the study. This process improved both the accuracy of policy-text identification and the validity of subsequent word-frequency analysis.

Second, word-frequency analysis was performed on the processed corpus to identify the most frequently occurring substantive terms. Expressions with limited analytical value, such as "shall not," "process," and "should," were excluded during preprocessing. The twenty most frequent terms in China's AI policy texts were then retained to construct the high-frequency word list (see Fig 4).

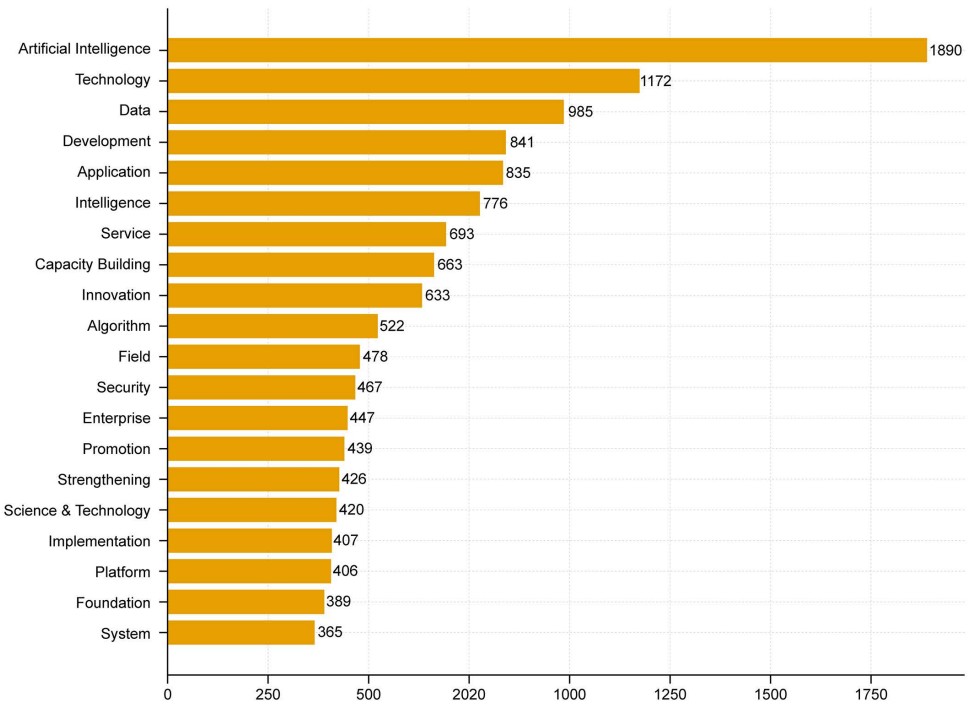

**Fig 4. Top 20 high-frequency terms in China's artificial intelligence policy documents.**

Third, to provide a more intuitive representation of the internal structure and relational patterns within China's AI policy texts, semantic co-occurrence analysis was conducted and visualized using NetDraw. The segmented and filtered terms were used to generate a policy keyword co-occurrence network (see Fig 5). In the resulting network, lines between nodes represent co-occurrence relationships between keywords, while nodes with a higher number of connections exhibit greater degree centrality and therefore occupy a more prominent position in the overall policy discourse structure.

As shown in Fig 4, the word-frequency distribution of China's AI policy corpus indicates that terms such as "artificial intelligence," "technology," "data," "development," "application," and "intelligence" appear most frequently, followed by "service," "capacity building," "innovation," "platform," "system," "security," and "management." This pattern suggests that China's AI policy framework is organized around three relatively stable thematic dimensions—technological capability, scenario implementation, and institutional support—reflecting a consistent orientation toward linking innovation capacity with practical deployment and governance mechanisms.

The results presented in Fig 5 directly reflect the primary emphases of China's AI policy discourse. Structurally, the co-occurrence network is centered on the core keywords Technology, Artificial Intelligence, Application, Development, and Innovation, which exhibit the highest frequency and the widest range of semantic connections. These terms occupy the central hub of the network and form three relatively stable cohesive subgroups.

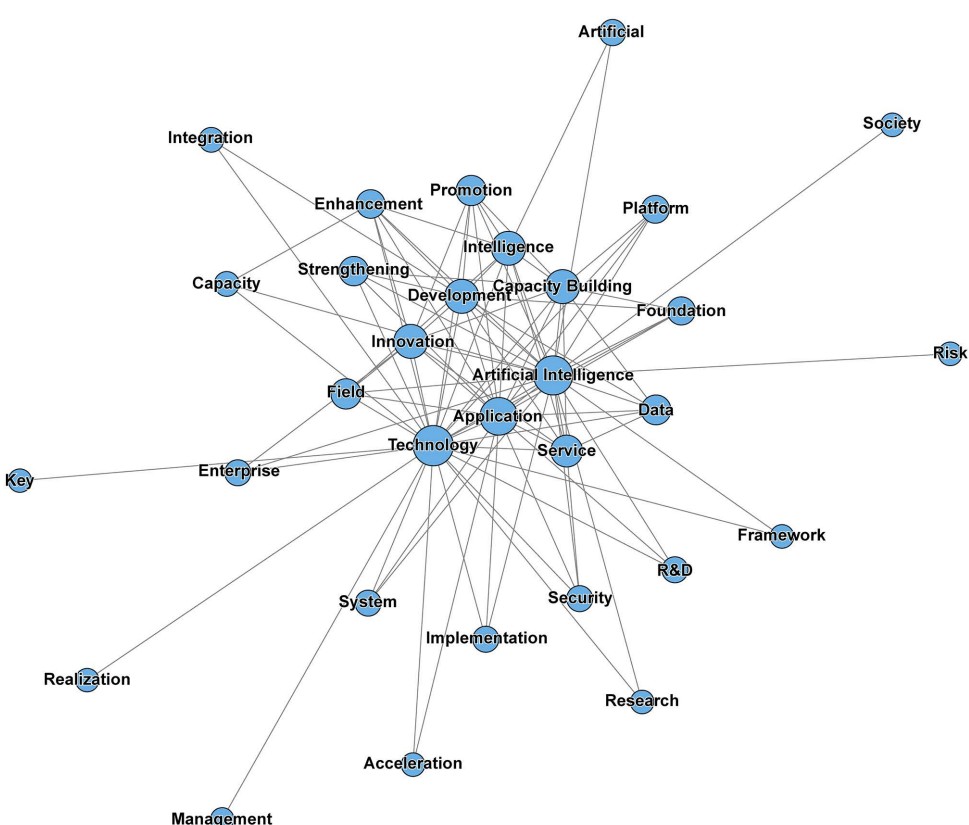

**Fig 5. Semantic co-occurrence network of high-frequency terms in China's artificial intelligence policy documents.**

From the perspective of content generation and thematic clustering, the semantic network can be interpreted as comprising three major clusters:

(1) The "technological foundation"cluster(Data–Platform–System–Foundation–CapacityBuilding–R&D/Research) demonstrates a high degree of connectivity, revealing China's integrative policy logic that links data, platforms, systems, basic research, and capacity building into a unified supply structure. The nodes platform and data play a prominent bridging role, connecting technological innovation with real-world application.

(2) The "governance and assurance" cluster (Security–Standard–Management, extending to Risk, Framework, and Evaluation) reflects the principle of "balancing development with security." This cluster highlights the regulatory boundaries of China's AI policy system, wherein standardization, management, and evaluation mechanisms are used to control potential risks within an institutional framework.

(3) The "application diffusion" cluster (Enterprise–Service–Field/Scenario) directly couples Application with Technology, pointing toward large-scale industrial deployment and the cultivation of public service ecosystems. Around this core, numerous action-oriented nodes—such as Implementation, Promotion, Strengthening, Enhancement, and Acceleration—are densely distributed, revealing an execution-oriented policy tendency driven by organizational advancement, scheduling, and capacity enhancement.

At the outer periphery of the network lie relatively marginal terms such as Risk, Society, and Management, which mainly carry conceptual or macro-level contextual meanings rather than operational functions. This core–belt–periphery structure indicates that China's AI policy network embodies a dual core of technology and application, bounded by the external constraints of capacity building as the foundation and security regulation as the boundary.

Based on these analytical findings, the study preliminarily identified several macro-level policy dimensions that informed the subsequent construction of the PMC variable system. These dimensions were not treated as automatically generated indicators, but were further refined through reference to prior policy-evaluation studies and structured interpretation of China's AI policy texts. They include:

• Strategic Orientation (Development,Innovation,Priority);

• Technology and R&D (Algorithm,Platform,System,Foundation,R&D);

• Data and Basic Capacity (Data,Standard,Capacity Building,Resource);

• Application and Industrialization (Application,Enterprise,Service,Field,Scenario);

• Governance,Security and Ethics (Security,Management,Framework,Risk,Evaluation,Ethics);

• Capacity Building and Implementation Mechanisms (Talent,Implementation,Coordination,Incentives,Strengthening,Enhancement).

**Variable classification and parameter identification.** In developing the hierarchical variables, this study drew upon three sources of evidence: relevant academic theories, existing empirical studies on policy evaluation, and close interpretation of China's AI policy texts. The text-mining results presented above were used to identify recurrent themes, semantic clusters, and policy emphases within the corpus, thereby providing an empirical basis for variable design. However, the indicator system was not treated as a direct output of automated text mining. Rather, it was constructed through a researcher-led process that combined corpus-based findings with prior policy-evaluation literature and structured interpretation of the policy texts.

On this basis, a total of nine primary variables and forty-three secondary variables were established to form the evaluation indicator system for the PMC model. The complete framework is presented in Table 2. Accordingly, Table 2 should be

Table 2. Quantitative evaluation indicator system and sources for China's artificial intelligence policiesprimary variable.

| | Secondary Variable | Evaluation Criteria | Source/ Reference | Analytical Function |
|---|---|---|---|---|
| **Policy Nature $X_1$** | Prediction($X_{1:1}$) | Indicates whether the policy includes predictive or foresight-oriented elements (1 = Yes; 0 = No) | Ruiz Estrada (2011) [25] | Reflects the government's foresight and regulatory orientation toward AI development, indicating the policy's directional, supportive, and regulatory functions in various development stages. |
| | Description($X_{1:2}$) | Describes the current status of development (1 = Yes; 0 = No) | | |
| | Recommendation($X_{1:3}$) | Includes explicit recommendations (1 = Yes; 0 = No) | | |
| | Guidance($X_{1:4}$) | Contains guiding provisions or instructions (1 = Yes; 0 = No) | | |
| | Regulation($X_{1:5}$) | Mentions supervision or regulatory mechanisms (1 = Yes; 0 = No) | | |
| | Support($X_{1:6}$) | Refers to support or assistance mechanisms (1 = Yes; 0 = No) | | |
| | Norms($X_{1:7}$) | Establishes normative or standard-setting elements (1 = Yes; 0 = No) | | |
| **Policy Timeliness $X_2$** | Long-term($X_{2:1}$) | Specifies policy content exceeding five years (1 = Yes; 0 = No) | Zhao (2023) [27] | Demonstrates the temporal scale and planning level of the policy's time horizon and target setting. |
| | Medium-term ($X_{2:2}$) | Specifies policy duration between three and five years (1 = Yes; 0 = No) | | |
| | Short-term ($X_{2:3}$) | Specifies policy duration less than three years (1 = Yes; 0 = No) | | |
| **Policy Level $X_3$** | State Council/ CPC Central Committee ($X_{3:1}$) | Indicates involvement of State Council or CPC Central Committee (1 = Yes; 0 = No) | Liu (2017) [28] | Reflects the authority of the policy-making body and the institutionalization of governance coordination mechanisms. |
| | Multiple Ministries($X_{3:2}$) | Jointly issued by multiple ministries (1 = Yes; 0 = No) | | |
| | Single Ministry ($X_{3:3}$) | Issued by one ministry only (1 = Yes; 0 = No) | | |
| **Policy Content $X_4$** | Platform Construction($X_{4:1}$) | Involves platform building (1 = Yes; 0 = No) | Derived from high-frequency term extraction and semantic network analysis based on ROSTCM6.0-assisted text mining of the policy corpus. | Reveals the core thematic focus and specific areas of policy attention. |
| | R&D and Technology ($X_{4:2}$) | Involves technological R&D (1 = Yes; 0 = No) | | |
| | Data Governance($X_{4:3}$) | Involves data governance (1 = Yes; 0 = No) | | |
| | Industrialization($X_{4:4}$) | Involves industrialization, enterprise participation, or scenario expansion (1 = Yes; 0 = No) | | |
| | Risk Governance($X_{4:5}$) | Refers to risk prevention or governance (1 = Yes; 0 = No) | | |
| | Scenario Application ($X_{4:6}$) | Involves application scenarios (1 = Yes; 0 = No) | | |
| **Policy Domain $X_5$** | Political ($X_{5:1}$) | Concerns political aspects (1 = Yes; 0 = No) | Gaofei Wang(2024) [29] | Indicates the principal fields of policy intervention. |
| | Economic($X_{5:2}$) | Concerns economic development (1 = Yes; 0 = No) | | |
| | Social($X_{5:3}$) | Concerns social dimensions (1 = Yes; 0 = No) | | |
| | Technological($X_{5:4}$) | Concerns technological innovation (1 = Yes; 0 = No) | | |
| | Institutional($X_{5:5}$) | Concerns institutional or regulatory structures (1 = Yes; 0 = No) | | |
| | Environmental ($X_{5:6}$) | Concerns ecological or environmental issues (1 = Yes; 0 = No) | | |

*(Continued)*

| | Secondary Variable | Evaluation Criteria | Source/ Reference | Analytical Function |
|---|---|---|---|---|
| **Incentive Measures $X_6$** | Government Subsidy ($X_{6:1}$) | Includes government subsidies (1 = Yes; 0 = No) | Shengli Dai(2021) [30] | Summarizes the policy tools and pathways supporting AI industry development. |
| | Tax Incentives ($X_{6:2}$) | Includes tax reduction or exemption measures (1 = Yes; 0 = No) | | |
| | Public Services($X_{6:3}$) | Refers to public service support (1 = Yes; 0 = No) | | |
| | Talent Attraction and Training ($X_{6:4}$) | Mentions talent introduction or training programs (1 = Yes; 0 = No) | | |
| | Intellectual Property Protection($X_{6:5}$) | Includes IP protection mechanisms (1 = Yes; 0 = No) | | |
| | Pilot and Demonstration Projects ($X_{6:6}$) | Mentions pilot projects or demonstration applications (1 = Yes; 0 = No) | | |
| **Policy Evaluation $X_7$** | Clear Objectives($X_{7:1}$) | States explicit policy objectives (1 = Yes; 0 = No) | Jing Gong(2025) [31] | Reflects the logical structure and practical feasibility of the policy text. |
| | Scientific Planning($X_{7:2}$) | Indicates whether the planning is scientifically sound (1 = Yes; 0 = No) | | |
| | Defined Responsibilities($X_{7:3}$) | Specifies clear division of responsibilities (1 = Yes; 0 = No) | | |
| | Evidence-Based Foundation($X_{7:4}$) | Indicates whether strategies are grounded in realistic conditions (1 = Yes; 0 = No) | | |
| **Target Actors $X_8$** | Government($X_{8:1}$) | Targets governmental institutions (1 = Yes; 0 = No) | Shuigen Hu(2025) [32] | Explains the functional mechanisms of AI policy among different governance actors. |
| | Enterprises($X_{8:2}$) | Targets enterprises (1 = Yes; 0 = No) | | |
| | Public ($X_{8:3}$) | Targets the general public (1 = Yes; 0 = No) | | |
| | Research Institutes($X_{8:4}$) | Targets academic or research institutions (1 = Yes; 0 = No) | | |
| **Level of Influence $X_9$** | National Innovation($X_{9:1}$) | Involves national innovation strategies (1 = Yes; 0 = No) | Yiwen Liu(2022) [33] | Reflects the spatial and organizational level at which policies exert influence. |
| | Regional Innovation($X_{9:2}$) | Involves regional innovation activities (1 = Yes; 0 = No) | | |
| | Industrial Development($X_{9:3}$) | Involves industrial development (1 = Yes; 0 = No) | | |
| | Enterprise Development($X_{9:4}$) | Involves enterprise capacity building (1 = Yes; 0 = No) | | |

understood as a researcher-constructed indicator system informed by both text analysis and theoretical reference, rather than as an automatically generated classification scheme. In substantive terms, the indicator design seeks to capture the multidimensional structure of AI policy, including policy nature, timeliness, policy level, policy content, policy domain, incentive measures, policy evaluation, target actors, and level of influence.

Following the construction of the indicator system, each secondary variable was operationalized using binary assignment criteria. A value of 1 was assigned when the corresponding policy element was explicitly present in the policy text, and 0 otherwise. This coding procedure enabled the transformation of qualitative policy content into a structured input format suitable for PMC evaluation. For transparency and reproducibility, the detailed indicator scoring matrix, including

evaluation criteria, operational coding notes, binary scoring rules, and source bases for all secondary variables, is provided in S3 File in S1 Data.

**Construction of the multi-input–output table.** The multi-input–output table operationalizes the indicator system presented in Table 2 by reorganizing the nine primary variables and their corresponding forty-three secondary variables into a standardized structure for PMC coding and calculation. In other words, Table 3 should be understood as the formal coding framework derived from Table 2, rather than as an independently generated analytical result.

Within this structure, each secondary variable is assigned a binary value according to the coding criteria defined in the indicator system. If a given policy text explicitly contains the policy element represented by a secondary variable, the value is coded as 1; otherwise, it is coded as 0. This binary assignment procedure allows the qualitative content of policy texts to be transformed into a structured and comparable input format for PMC evaluation. Based on this logic, the multi-input–output table constructed for China's AI policies is presented in Table 3.

**Calculation of the PMC index.** The calculation of the Policy Modeling Consistency (PMC) Index proceeded on the basis of the standardized coding structure shown in Table 3. First, the binary values assigned to all secondary variables were entered into the multi-input–output table. Second, the values of the primary variables were calculated according to the aggregation rules specified in Equations (1)–(3). Third, the PMC Index of each policy was obtained by summing the values of all primary variables in accordance with Equation (4).

In this process, the role of Table 4 is interpretive rather than computational. Specifically, Table 4 provides the policy classification standard used to interpret the PMC scores after they have been calculated. Referring to previous studies [34], this study adopts four levels of PMC-based policy evaluation, ranging from "Poor" to "Excellent." Accordingly, Table 4 should be understood as the analytical basis for score interpretation, rather than as a table generated directly from text-mining output or coding procedures.

**Table 3. Multi-input–output table of PMC model variables for China's artificial intelligence policies.**

| Primary Variable | Secondary Variable |
|---|---|
| $X_1$ | $X_{1:1}, X_{1:2}, X_{1:3}, X_{1:4}, X_{1:5}, X_{1:6}, X_{1:7}$ |
| $X_2$ | $X_{2:1}, X_{2:2}, X_{2:3}$ |
| $X_3$ | $X_{3:1}, X_{3:2}, X_{3:3}$ |
| $X_4$ | $X_{4:1}, X_{4:2}, X_{4:3}, X_{4:4}, X_{4:5}, X_{4:6}$ |
| $X_5$ | $X_{5:1}, X_{5:2}, X_{5:3}, X_{5:4}, X_{5:5}, X_{5:6}$ |
| $X_6$ | $X_{6:1}, X_{6:2}, X_{6:3}, X_{6:4}, X_{6:5}, X_{6:6}$ |
| $X_7$ | $X_{7:1}, X_{7:2}, X_{7:3}, X_{7:4}$ |
| $X_8$ | $X_{8:1}, X_{8:2}, X_{8:3}, X_{8:4}$ |
| $X_9$ | $X_{9:1}, X_{9:2}, X_{9:3}, X_{9:4}$ |

**Table 4. PMC index–based policy classification standards.**

| PMC index range | Policy evaluation level |
|---|---|
| [8,10] | Excellent |
| [7,8] | Good |
| [4,7] | Qualified |
| [0,3] | Poor |

The calculation of the PMC Index therefore involved two linked stages: the transformation of coded policy content into measurable index values, and the interpretation of those values according to the PMC classification standard. The specific equations used in this study are presented below.

$$X : N[0, 1] \tag{1}$$

$$X = \{XR : [0, 1]\} \tag{2}$$

$$X_t \left( \sum_{j=1}^{n} \frac{X_{tj}}{T(X_{tj})} \right) \tag{3}$$

$$\text{PMC} = X_1 \left( \frac{1}{7} \sum_{j=1}^{7} X_{1j} \right) + X_2 \left( \frac{1}{3} \sum_{j=1}^{3} X_{2j} \right) + X_3 \left( \frac{1}{3} \sum_{j=1}^{3} X_{3j} \right) + X_4 \left( \frac{1}{6} \sum_{j=1}^{6} X_{4j} \right) + X_5 \left( \frac{1}{6} \sum_{j=1}^{6} X_{5j} \right)$$
$$+ X_6 \left( \frac{1}{6} \sum_{j=1}^{6} X_{6j} \right) + X_7 \left( \frac{1}{4} \sum_{j=1}^{4} X_{7j} \right) + X_8 \left( \frac{1}{4} \sum_{j=1}^{4} X_{8j} \right) + X_9 \left( \frac{1}{4} \sum_{j=1}^{4} X_{9j} \right). \tag{4}$$

To facilitate substantive interpretation, the PMC Index in this study is understood as an indicator of the structural consistency of policy design across the selected evaluation dimensions. A higher PMC score does not imply that a policy is universally superior in every respect; rather, it indicates that the policy exhibits a relatively more complete, balanced, and internally coordinated configuration across dimensions such as policy nature, timeliness, content, incentive measures, target actors, and level of influence. By contrast, a lower PMC score suggests that, although the policy may still be valid and practically relevant, its structural design is relatively less comprehensive or less balanced, with one or more dimensions receiving weaker coverage or support.

In practical terms, relatively strong structural consistency refers to a policy design in which strategic orientation, temporal planning, substantive content, implementation support, and governance targets are aligned in a comparatively coherent manner. Relatively weak structural consistency, by contrast, refers to a policy structure in which one or more of these dimensions are underdeveloped, weakly connected, or insufficiently supported, thereby limiting the overall internal coherence of the policy. Accordingly, policies classified as "Excellent" can be interpreted as exhibiting a comparatively high level of structural completeness and coordination across major dimensions; those classified as "Good" indicate a generally coherent policy structure with some remaining weaker aspects; those classified as "Qualified" reflect a more basic or uneven structural configuration; and those classified as "Poor" suggest a relatively fragmented or weakly supported policy structure.

**PMC surface equation.** To present the PMC Index in a more intuitive three-dimensional form for facilitating policy evaluation and improvement, the PMC surface visualization in this study was implemented using Python (version 3.8). The Matplotlib and SciPy libraries were employed to generate three-dimensional plots.

In this model, nine primary variables were established, forming a 3 × 3 PMC matrix, and the calculation of the PMC surface is expressed as shown in Equation (5) below:

$$PMC\ surface\ matrix = \begin{bmatrix} X_1 & X_2 & X_3 \\ X_4 & X_5 & X_6 \\ X_7 & X_8 & X_9 \end{bmatrix} \tag{5}$$

## Transparency and reproducibility materials

To enhance methodological transparency and reproducibility, this study provides supplementary materials documenting the core steps of the analytical process. These materials cover corpus construction, text processing, indicator operationalization, and PMC scoring, making the analytical procedure explicit and verifiable. Because the main manuscript prioritizes analytical clarity and representative findings, detailed supporting documentation is provided separately in the supplementary files.

First, the supplement includes a complete list of the 54 central-level AI policy documents comprising the corpus, along with essential metadata (e.g., policy title, issuance year, issuing authority, and source). While the main text highlights selected representative policies for readability, the full corpus is provided to support sample verification and future comparative research.

Second, to demystify the text-mining stage, the text-processing notes and the segmentation-related lexical resource used in the text-mining stage are included. These materials clarify how raw policy texts were segmented, filtered, and matched to relevant policy features, thereby improving the interpretability of the transition from raw data to analytical indicators.

Third, the files detail the operational scoring matrix for the PMC evaluation system. This encompasses the definitions of the nine primary variables and 43 secondary indicators, the binary scoring criteria for each, and the coding notes applied during the assignment process. Providing this information ensures the scoring logic remains transparent and minimizes ambiguity in indicator interpretation.

Finally, the complete PMC evaluation outputs are reported in the supplement. Beyond the representative findings discussed in the main text, providing the full set of scoring results facilitates independent verification and methodological replication.

Taken together, these materials extend the study's "text–indicator–metric–visualization" pipeline, establishing a robust foundation for verification, replication, and future cross-jurisdictional comparisons in AI policy research.

## Empirical evaluation of China's artificial intelligence policies

### Selection of evaluation objects

The full corpus of 54 central-level AI policy documents was used for text mining and variable identification, whereas five representative policy documents were selected for detailed PMC evaluation. The evaluation period for these representative policy samples corresponds to China's 14th Five-Year Plan period. This period was chosen because it represents a comparatively mature and policy-intensive stage in the recent development of China's AI governance framework, during which AI policy became more systematically linked to national strategies of digital transformation, industrial upgrading, and governance modernization. The specific information on the five selected policies is presented in Table 5.

The selection of evaluation objects was based on the following considerations.

First, policy level. This study focuses on AI policies issued at the national level of China. Accordingly, all five selected cases are central-level policy documents, ensuring consistency in governance hierarchy and institutional authority.

Second, policy issuance time. The selected documents were all issued during the 14th Five-Year Plan period (2021–2025), which ensures temporal continuity and reflects the most recent stage of central-level AI policy development.

Third, policy type and structural comparability. The selected documents mainly take the form of opinions, action plans, and implementation-oriented policy measures aimed at promoting, regulating, or coordinating AI development. Their relatively comparable policy structure makes them suitable for PMC-based evaluation of internal consistency.

Fourth, representativeness. The selected policies differ in issuing authority, policy emphasis, and governance orientation, thereby enabling a comparative analysis of how structural consistency varies across different types of central-level AI policy arrangements within the same broader policy cycle.

**Table 5. Selected AI policy samples in China.**

| Code | Policy Title | Issuing Authority | Date of Issue |
|------|--------------|-------------------|---------------|
| P1 | Guiding Opinions on Accelerating Scenario-Based Innovation to Promote High-Level Application of Artificial Intelligence and High-Quality Economic Development | Ministry of Science and Technology, Ministry of Education, Ministry of Industry and Information Technology, Ministry of Transport, Ministry of Agriculture and Rural Affairs, National Health Commission | Jul-22 |
| P2 | Notice on Supporting the Construction of New Generation Artificial Intelligence Demonstration Application Scenarios | Ministry of Science and Technology | Aug-22 |
| P3 | Interim Measures for the Administration of Generative Artificial Intelligence Services | Cyberspace Administration of China; National Development and Reform Commission; Ministry of Education; Ministry of Science and Technology; Ministry of Industry and Information Technology; Ministry of Public Security; National Radio and Television Administration | Aug-23 |
| P4 | Notice on Strengthening Artificial Intelligence Education in Primary and Secondary Schools | Ministry of Education | Aug-24 |
| P5 | Opinions on Deeply Implementing the "AI+" Action Plan | State Council | Aug-25 |

## Construction of the multi-input–output table

Based on the PMC indicator system established in Section 3, the multi-input–output table for the selected policy cases was constructed using nine primary variables and forty-three secondary variables. The purpose of this step was to translate the indicator framework into a standardized coding structure for subsequent PMC calculation.

More specifically, the coding process was carried out on the basis of the predefined binary scoring matrix presented in the revised Methods section and the corresponding Supporting Information. The identification of each secondary variable was informed by the policy texts themselves, together with the analytical dimensions derived from the preceding text-mining stage and relevant policy-evaluation literature. In this process, the text-mining results served as an empirical reference for identifying salient policy dimensions, whereas the final assignment of variable values relied on structured researcher judgment according to explicit coding criteria.

For each selected policy, a value of 1 was assigned when the corresponding policy element was explicitly present in the text, and 0 otherwise. This procedure transformed qualitative policy content into a structured input format that could be used for PMC evaluation. On this basis, the assigned values of the secondary variables for the five selected AI policies were organized into the multi-input–output table shown in Table 6, which provided the direct basis for subsequent PMC Index calculation and visualization.

## Calculation of the PMC index

Based on the procedures described above, and following the establishment of the multi-input–output table, the PMC Index and corresponding mean values for each policy were calculated according to Equations (3) and (4). Subsequently, each policy was classified into its respective level based on the policy classification standards defined earlier. The detailed results are presented in Table 7.

Accordingly, differences in PMC scores should not be understood merely as numerical ranking differences, but as reflecting variation in the completeness and internal coordination of policy design. In this study, a higher PMC score indicates that the policy is configured more comprehensively and coherently across the selected dimensions, including policy nature, timeliness, content, incentive measures, target actors, and level of influence. By contrast, a lower PMC score suggests that the policy structure is relatively less balanced or less fully developed, even if the policy remains valid and relevant in practice.

**Table 6. Multi-input–output table for the five selected AI policies in China.**

| Primary Variable | Secondary Variable | P1 | P2 | P3 | P4 | P5 |
|---|---|---|---|---|---|---|
| Policy Nature $X_1$ | Prediction($X_{1:1}$) | 1 | 1 | 1 | 1 | 1 |
| | Description($X_{1:2}$) | 1 | 1 | 1 | 1 | 1 |
| | Recommendation($X_{1:3}$) | 1 | 0 | 1 | 1 | 1 |
| | Guidance($X_{1:4}$) | 1 | 1 | 1 | 1 | 1 |
| | Regulation($X_{1:5}$) | 0 | 0 | 1 | 0 | 1 |
| | Support($X_{1:6}$) | 1 | 1 | 1 | 1 | 1 |
| | Norms($X_{1:7}$) | 1 | 0 | 1 | 1 | 1 |
| Policy Timeliness $X_2$ | Long-term($X_{2:1}$) | 0 | 0 | 0 | 1 | 1 |
| | Medium-term($X_{2:2}$) | 0 | 0 | 0 | 0 | 1 |
| | Short-term ($X_{2:3}$) | 1 | 1 | 1 | 0 | 1 |
| Policy Level $X_3$ | State Council/ CPC Central Committee ($X_{3:1}$) | 0 | 0 | 0 | 0 | 1 |
| | Multiple Ministries($X_{3:2}$) | 1 | 0 | 1 | 0 | 0 |
| | Single Ministry($X_{3:3}$) | 0 | 1 | 0 | 1 | 0 |
| Policy Content $X_4$ | Platform Construction($X_{4:1}$) | 1 | 1 | 1 | 0 | 1 |
| | R&D and Technology ($X_{4:2}$) | 1 | 1 | 1 | 0 | 1 |
| | Data Governance($X_{4:3}$) | 1 | 1 | 1 | 0 | 1 |
| | Industrialization($X_{4:4}$) | 1 | 1 | 0 | 0 | 1 |
| | Risk Governance($X_{4:5}$) | 0 | 1 | 1 | 0 | 1 |
| | Scenario Application($X_{4:6}$) | 1 | 1 | 0 | 1 | 1 |
| Policy Domain $X_5$ | Political ($X_{5:1}$) | 1 | 1 | 1 | 1 | 1 |
| | Economic($X_{5:2}$) | 1 | 1 | 0 | 0 | 1 |
| | Social($X_{5:3}$) | 1 | 1 | 1 | 1 | 1 |
| | Technological($X_{5:4}$) | 1 | 1 | 1 | 0 | 1 |
| | Institutional($X_{5:5}$) | 1 | 0 | 1 | 1 | 1 |
| | Environmental ($X_{5:6}$) | 1 | 1 | 1 | 1 | 1 |
| Incentive Measures $X_6$ | Government Subsidy($X_{6:1}$) | 1 | 0 | 0 | 1 | 1 |
| | Tax Incentives ($X_{6:2}$) | 1 | 0 | 0 | 0 | 1 |
| | Public Services($X_{6:3}$) | 1 | 1 | 1 | 1 | 1 |
| | Talent Attraction and Training($X_{6:4}$) | 1 | 0 | 0 | 1 | 1 |
| | Intellectual Property Protection($X_{6:5}$) | 0 | 0 | 1 | 0 | 1 |
| | Pilot and Demonstration Projects($X_{6:6}$) | 1 | 1 | 0 | 1 | 1 |
| Policy Evaluation $X_7$ | Clear Objectives($X_{7:1}$) | 0 | 1 | 1 | 1 | 1 |
| | Scientific Planning($X_{7:2}$) | 0 | 1 | 1 | 1 | 1 |
| | Defined Responsibilities($X_{7:3}$) | 0 | 1 | 1 | 1 | 1 |
| | Evidence-Based Foundation($X_{7:4}$) | 1 | 1 | 1 | 1 | 1 |
| Target Actors $X_8$ | Government($X_{8:1}$) | 1 | 0 | 1 | 1 | 1 |
| | Enterprises($X_{8:2}$) | 1 | 1 | 1 | 0 | 1 |
| | Public ($X_{8:3}$) | 1 | 1 | 1 | 1 | 1 |
| | Research Institutes($X_{8:4}$) | 1 | 0 | 1 | 1 | 1 |
| Level of Influence $X_9$ | National Innovation($X_{9:1}$) | 1 | 1 | 0 | 0 | 1 |
| | Regional Innovation($X_{9:2}$) | 1 | 0 | 0 | 0 | 1 |
| | Industrial Development($X_{9:3}$) | 1 | 1 | 1 | 0 | 1 |
| | Enterprise Development($X_{9:4}$) | 1 | 1 | 1 | 0 | 1 |

**Table 7. PMC index of the five selected AI Policies in China.**

| Primary Variable | P1 | P2 | P3 | P4 | P5 | Mean |
|---|---|---|---|---|---|---|
| Policy Nature $X_1$ | 1 | 0.86 | 1 | 0.86 | 1 | 0.944 |
| Policy Timeliness $X_2$ | 0.33 | 0.33 | 0.33 | 0.33 | 1 | 0.464 |
| Policy Level $X_3$ | 0.33 | 0.33 | 0.33 | 0.33 | 0.33 | 0.33 |
| Policy Content $X_4$ | 0.83 | 0.83 | 0.67 | 0.33 | 1 | 0.732 |
| Policy Domain $X_5$ | 1 | 0.83 | 0.83 | 0.83 | 1 | 0.898 |
| Incentive Measures $X_6$ | 0.67 | 0.33 | 0.5 | 0.67 | 0.83 | 0.6 |
| Policy Evaluation $X_7$ | 0.25 | 1 | 1 | 1 | 1 | 0.85 |
| Target Actors $X_8$ | 1 | 0.5 | 1 | 0.75 | 1 | 0.85 |
| Level of Influence $X_9$ | 1 | 0.25 | 0.5 | 0 | 1 | 0.55 |
| PMC Index | 7.1222 | 5.8444 | 6.8444 | 5.6667 | 9.0667 | 6.909 |
| Ranking | 2 | 4 | 3 | 5 | 1 | — |
| Evaluation Grade | Good | Qualified | Qualified | Qualified | Excellent | Qualified |

In practical terms, relatively strong structural consistency refers to a policy arrangement in which strategic orientation, temporal planning, substantive content, implementation support, and governance targets are aligned in a comparatively coherent and mutually reinforcing manner. Relatively weak structural consistency, by contrast, refers to a policy structure in which one or more of these dimensions are underdeveloped, weakly connected, or insufficiently supported, thereby reducing the overall internal coherence of the policy.

This interpretive framework helps explain the differences among the five representative policies. For example, P5 received the highest PMC score because it shows a comparatively complete and coordinated structure across multiple dimensions, especially in policy timeliness, content coverage, incentive measures, target actors, and level of influence. By contrast, lower-scoring policies such as P4 reveal a more uneven structural configuration, with relatively limited performance in dimensions such as timeliness, content completeness, and influence scope. Therefore, the PMC results should be interpreted not only as quantitative rankings, but also as indicators of the relative structural completeness and coordination of AI policy design.

**PMC surface and radar chart visualization.** The PMC surface provides a three-dimensional representation of the structural profile of each policy, allowing differences across the nine primary indicators to be visualized more intuitively. To generate the PMC surface, the normalized scores of the nine primary variables were first arranged into a 3 × 3 PMC matrix for each representative policy sample (see Table 8). Based on these matrices, the PMC surfaces were then plotted for P1–P5 (see Figs 6–10). In the resulting surface graphs, elevated areas indicate relatively stronger dimensions of policy design, whereas concave areas indicate dimensions with comparatively lower scores. This visualization makes it possible

**Table 8. PMC matrices of the five selected AI policies in China.**

| Policy Item | P1 | P2 | P3 |
|---|---|---|---|
| PMC Matrix | $\begin{pmatrix} 1 & 0.33 & 0.33 \\ 0.83 & 1 & 0.67 \\ 0.25 & 1 & 1 \end{pmatrix}$ | $\begin{pmatrix} 0.86 & 0.33 & 0.33 \\ 0.83 & 0.83 & 0.33 \\ 1 & 0.50 & 0.25 \end{pmatrix}$ | $\begin{pmatrix} 1 & 0.33 & 0.33 \\ 0.67 & 0.83 & 0.50 \\ 1 & 1 & 0.50 \end{pmatrix}$ |

| Policy Item | P4 | P5 |
|---|---|---|
| PMC Matrix | $\begin{pmatrix} 0.86 & 0.33 & 0.33 \\ 0.33 & 0.83 & 0.67 \\ 1 & 0.75 & 0 \end{pmatrix}$ | $\begin{pmatrix} 1 & 1 & 0.33 \\ 1 & 1 & 0.83 \\ 1 & 1 & 1 \end{pmatrix}$ |

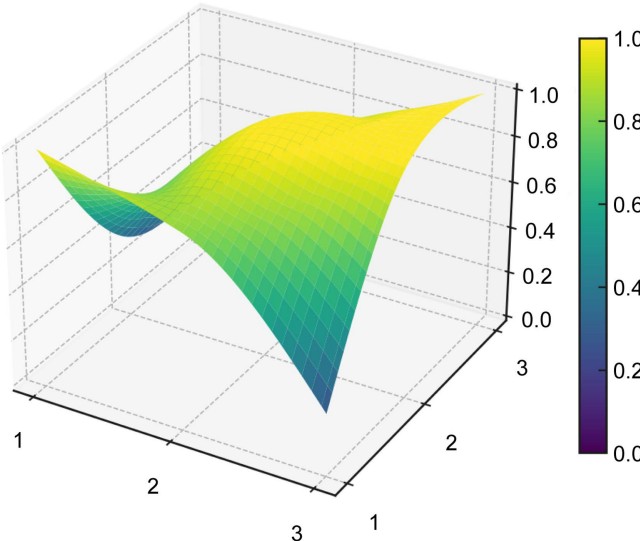

**Fig 6. PMC surface of policy P1.**

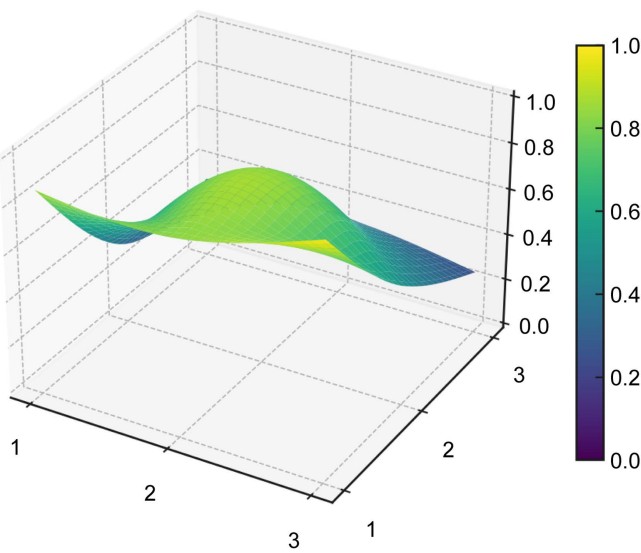

**Fig 7. PMC surface of policy P2.**

to identify not only overall policy quality, but also the internal distribution of strengths and weaknesses across policy dimensions.

The PMC surface visualization in this study was implemented in Python (version 3.8) using the Matplotlib and SciPy libraries. Specifically, the 3 × 3 matrices of primary-variable scores were smoothed through interpolation prior to three-dimensional plotting, so that the resulting surfaces would more clearly display structural variation across dimensions. In this representation, the x- and y-axes indicate matrix coordinates, while the z-axis represents the normalized scores of the PMC primary variables.

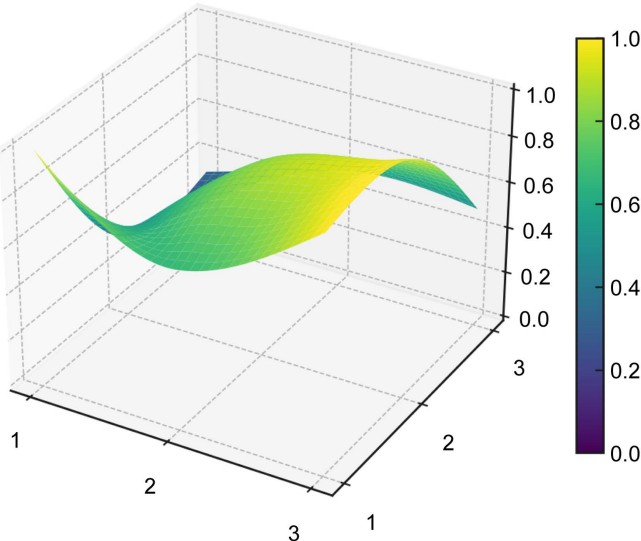

**Fig 8. PMC surface of policy P3.**

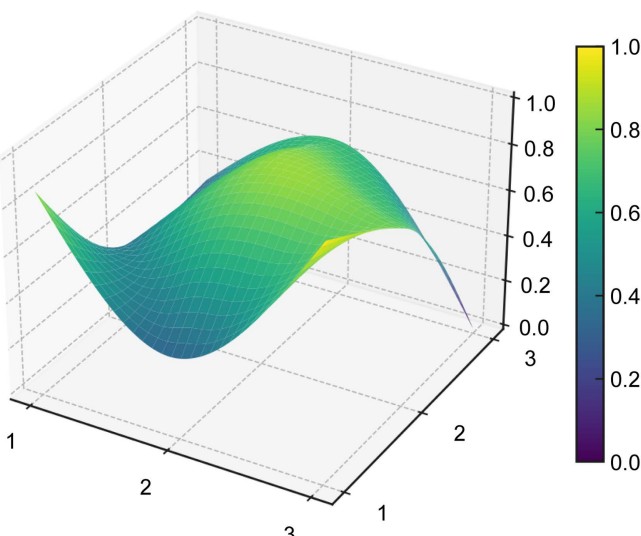

**Fig 9. PMC surface of policy P4.**

To complement the PMC surfaces, radar charts were also generated in order to display each policy's performance across the nine primary indicators ($X_1$–$X_9$)in a two-dimensional comparative form. Using the normalized scores of the primary indicators as radial values, the profiles of P1–P5 were plotted sequentially to form nine-dimensional contours. In addition, the sample mean curve (shown as a dashed line) was included as a baseline for comparison, making it easier to observe the relative dispersion and concentration of policy performance across dimensions. Compared with the PMC

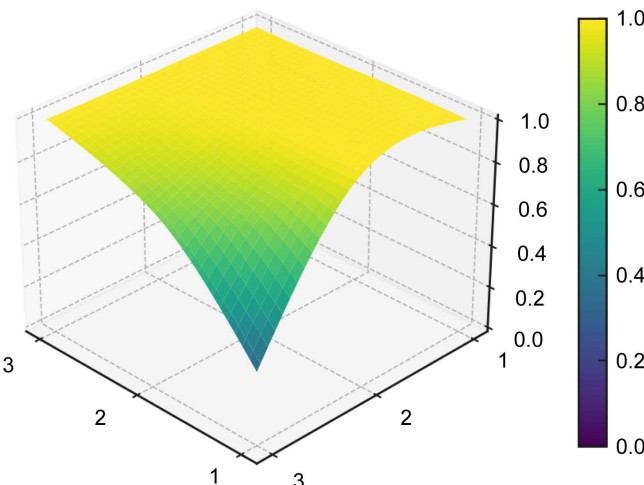

**Fig 10. PMC surface of policy P5.**

surface, the radar chart provides a more concise representation of cross-policy differences in structural configuration. The results are presented in Fig 11.

### Empirical results and analysis

**Overall quality assessment of China's AI policies.** Based on the PMC indicator system and assigned values derived from the five representative policies (see Table 7), the overall quality of China's AI policy system is relatively high

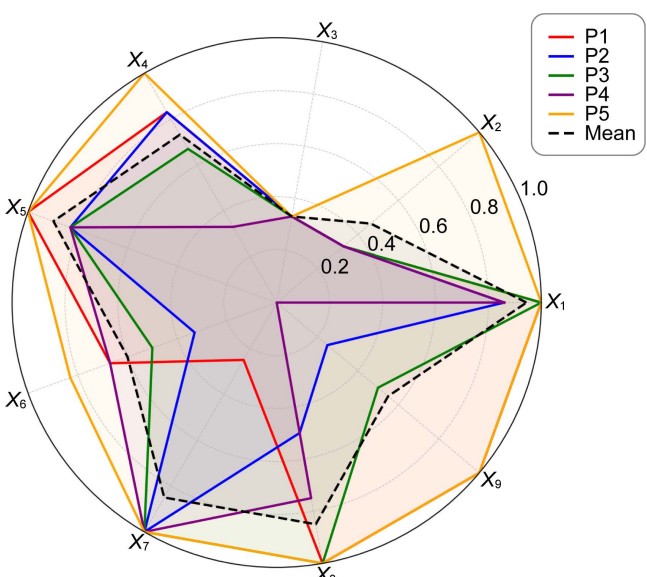

**Fig 11. Radar chart of China's artificial intelligence policies.**

but exhibits structural imbalance. The PMC Index of the five policies ranges from 5.67 to 9.07, with an average of 6.91, corresponding to the evaluation levels of Qualified–Good–Excellent. From a horizontal comparison, Policy P5 (Opinions on Deeply Implementing the "AI+" Action Plan, State Council, 2025) ranks first with a score of 9.07, demonstrating the comprehensive advantages of top-level strategic documents in terms of strategic coherence, implementation pathways, and actor coverage. In contrast, Policy P4 (Notice on Strengthening Artificial Intelligence Education in Primary and Secondary Schools, Ministry of Education, 2024) scores 5.67, indicating that specialized education policies remain limited in cross-sectoral coordination, hierarchical depth, and long-term planning.

Further analysis incorporating the radar chart (Figure 11), which includes the P1–P5 results and the mean curve, reveals the relative positioning of the nine primary indicators. $X_1$ (Policy Nature) and $X_5$ (Policy Domain) consistently extend outward across all policies, with mean values of 0.944 and 0.898, respectively. This indicates that the sampled policies generally possess clear strategic positioning and comprehensive domain coverage—consistent with China's national strategy of integrating AI into the industrial system under the framework of new productive forces. In contrast, $X_2$ (Policy Timeliness) and $X_3$ (Policy Level) are more contracted, with mean values of 0.464 and 0.33, respectively. This reflects a general tendency toward short- and medium-term guidance dominated by ministerial-level policies, with limited cross-level coordination and long-cycle (five-year or longer) planning—a typical phenomenon in the transition from policy proliferation to institutional stabilization.

From a functional perspective, $X_4$ (Policy Content) and $X_7$ (Policy Evaluation), with mean values of 0.732 and 0.85, respectively, form the backbone of policy text quality. The former covers the main dimensions of platform construction, technological R&D, data governance, scenario application, industrialization, and risk management, while the latter demonstrates a structured orientation through clearly defined objectives, planning, responsibilities, and evidence-based foundations. In the radar chart, P5 exhibits outwardly extended axes across $X_1$, $X_4$, $X_5$, $X_6$, $X_8$, and $X_9$, and its PMC surface (see Figs 6–10) shows multiple elevated points, reflecting a policy that combines strategic clarity, tool sufficiency, and implementation orientation. Conversely, P4 shows notable contraction in $X_2$ and $X_9$, indicating insufficient "terminal adhesion" in sustaining long-term governance and multi-level coordination.

The mean value of $X_6$ (Incentive Measures) is 0.60, showing the greatest variation among policies. The outward extension of P5 and P1 demonstrates a comprehensive configuration of fiscal subsidies, public services, pilot projects, and talent initiatives, while P2 and P3 remain conservative in applying more actionable tools such as tax incentives and intellectual property protection, suggesting that industrial implementation still requires a stronger mix of incentives. The overall fullness of $X_8$ (Target Actors) (mean = 0.85) reflects a well-established multi-actor framework encompassing government, enterprises, research institutions, and the public. However, $X_9$ (Level of Influence), with a mean value of 0.55, shows an inward contraction, revealing gaps in the four-level linkage among national, regional, industrial, and enterprise domains—particularly in the transmission mechanisms bridging national strategy and enterprise-level execution.

In summary, the empirical results reveal a "dual characteristic" of China's AI policy quality performance. On one hand, guided by top-level strategic documents, the policy system demonstrates strong strategic orientation and domain coverage; on the other hand, policies issued by different departments and administrative levels vary significantly in timeliness, incentives, and hierarchical coordination. This indicates that China's AI policy framework is still in a process of optimization and refinement—showing assertive progress in strategic design while suggesting the need for stronger long-term orientation, enhanced cross-departmental coordination, and more detailed incentive mechanisms to build a truly multi-level and full-chain AI policy system.

**Analysis of the distribution patterns of andicators in China's AI policies.** To compare the differences in the performance of various primary indicators among the sample policies and to examine potential outlier samples, this study plotted boxplots for the nine primary indicators based on the assigned data of P1–P5 (see Fig 12). Overall, the indicator distribution presents a structural pattern characterized by "strategic and domain stability, timeliness and hierarchy dispersion, and implementation–incentive differentiation."

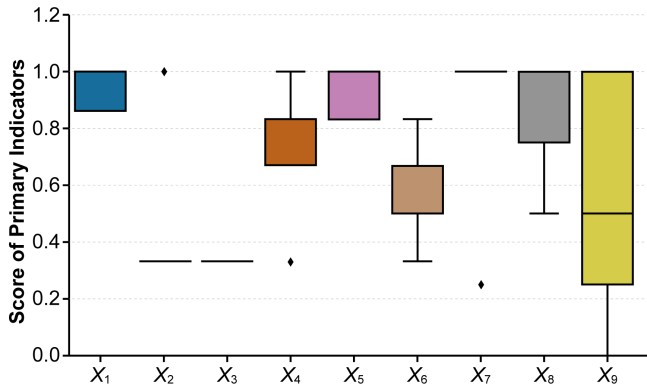

**Fig 12. Distribution of nine primary indicators across five representative Chinese artificial intelligence policies.** (Note: The black dots in the figure represent outliers in each data group.).

First, the boxplots for $X_1$ (Policy Nature) and $X_5$ (Policy Domain) are tightly clustered, with small interquartile ranges and overall high scores, and few outliers. This indicates a strong consensus among the sampled policies in terms of strategic orientation and domain coverage. Most policies combine directive, advisory, and normative functions, and systematically encompass key domains such as economy, society, technology, institutions, and environment. This aligns with China's positioning of artificial intelligence as a core driver within the "new quality productive forces–industrial system upgrading" framework, demonstrating that the top-level design of AI policies has reached a relatively mature stage regarding domain boundaries and functional definitions. Conversely, $X_3$ (Policy Level) shows almost no variance, concentrated at the ministerial or multi-ministerial level (0.33), reflecting that during the sampling period, few overarching central documents were issued. Most policy implementation relied on departmental regulations, explaining the observed insufficiency in cross-level coordination and vertical policy transmission.

Second, $X_2$ (Policy Timeliness) and $X_9$ (Level of Influence) display the highest degree of dispersion. The box for $X_2$ is overall low (mainly short- and medium-term), but includes an upper outlier of 1.0 corresponding to the State Council's P5, which features long-term planning. This suggests that China's current AI policies still rely primarily on phased promotion rather than systematic long-term planning. The range for $X_9$ spans 0–1, revealing significant disparities across the "national–regional–industrial–enterprise" transmission chain: P5 and P1 demonstrate strong alignment from national strategies to industrial and enterprise-level implementation, whereas P4, focused on educational scenarios, exhibits significant contraction in this dimension. This indicates that specialized policies lack institutionalized linkage to downstream actors—a finding consistent with the PMC surface and radar chart patterns discussed earlier.

Third, $X_4$ (Policy Content) and $X_7$ (Policy Evaluation) show a "strong overall performance with isolated outliers" structure. The median of $X_4$ is relatively high, but a low-end outlier (0.33, P4) suggests that when policies focus on a single scenario, they tend to exhibit uneven coverage across the full chain of "platform–R&D–data–application–industrialization–risk governance." $X_7$ remains at a high overall level, reflecting a general trend toward normalization in goal-setting, planning, accountability, and evidence bases. However, the lower outlier (0.25, P1) reveals that even inter-ministerial policies can experience quality reduction if they lack measurable goals, clear accountability lists, or solid evidence foundations.

Fourth, the boxplot for $X_6$ (Incentive Measures) lies in the midrange but features long whiskers, indicating heterogeneity in the combination strength of policy tools such as fiscal subsidies, tax incentives, intellectual property protection, pilot projects, and talent mechanisms. The "non-modular policy toolkit" problem thus persists. Meanwhile, $X_8$ (Target Actors) scores high overall but extends downward to 0.5, showing that while multi-stakeholder inclusion (government, enterprises,

academia, and the public) has become the norm, support intensity still varies notably across enterprises and research institutions.

In summary, the boxplot distribution provides three pieces of optimization evidence:

(1) Regarding the high dispersion of policy timeliness ($X_2$), a unified mid- to long-term schedule and rolling evaluation mechanism should be established between central and ministerial policies to avoid fragmented policy pacing.

(2) Concerning the wide range of influence levels ($X_9$), regional pilots and industrial platforms should be leveraged to improve the "national–regional–industrial–enterprise" feedback loop, enhancing terminal policy absorption and diffusion capacity.

(3) Addressing the outliers in content and evaluation ($X_4$, $X_7$) and the differentiation in incentives ($X_6$), integrated policy packages should be developed that jointly incorporate scenario demonstration, data governance, risk regulation, and talent mechanisms, grounded in verifiable goals, accountability lists, and evidence-based frameworks to build a replicable and traceable evaluation system.

These findings align with China's transition from intensive policy issuance to institutional stabilization in the AI domain and provide an actionable direction for future robustness tests and sensitivity analyses on larger policy samples.

## Discussion and implications

### Discussion

This study used the PMC Index Model to examine the structural consistency of China's central-level AI policies and yielded three principal findings. First, the policy system displays a relatively stable core orientation centered on technological development, application expansion, and governance support. Second, despite this overall thematic stability, important weaknesses remain in temporal planning and policy hierarchy, especially with regard to long-term policy design and cross-level coordination. Third, marked variation persists in incentive arrangements and influence levels, indicating that the internal configuration of China's AI policy system remains uneven across different dimensions of implementation.

The first finding reinforces and extends previous studies that have characterized AI policy as shifting from a narrow techno-economic agenda toward a broader governance-oriented framework [5,6,11]. Existing content-based and computational studies have shown that Chinese AI policies place strong emphasis on innovation, development, and security [13–15]. The present study confirms this general pattern, but moves beyond thematic description by showing that these concerns are not merely frequent topics in policy discourse; rather, they constitute a relatively stable structural core within China's central-level AI policy design. In this sense, our findings support prior observations regarding the developmental and governance-oriented character of China's AI policy system, while also demonstrating that this orientation is embedded in the internal configuration of policy elements rather than only in rhetorical emphasis.

The second finding suggests that the main limitations of China's central-level AI policies no longer lie primarily in the absence of policy content, but in the uneven structural integration of policy time horizons, governance hierarchy, and implementation linkage. This result is broadly consistent with earlier studies arguing that many national AI strategies suffer from insufficient implementation detail, limited evaluation mechanisms, and weak operational follow-through [8]. However, the present study adds a more differentiated structural diagnosis. Rather than treating implementation weakness as a general shortcoming, the PMC analysis shows more specifically that temporal planning remains concentrated in the short and medium term, while cross-level coordination is relatively underdeveloped. This indicates that current AI governance arrangements in China are not primarily constrained by thematic insufficiency, but by the incomplete institutionalization of long-range planning and vertical policy integration.

A third finding concerns the heterogeneity of incentive measures and influence levels. This result resonates with previous PMC-based policy studies showing that incentive mechanisms and coordination structures often constitute the most

uneven dimensions of policy design [19–24]. In the case of China's AI policy system, the present findings suggest that policy support is distributed asymmetrically across fiscal, tax, talent, service, and demonstration instruments, and that the transmission of policy influence across national, regional, industrial, and enterprise levels remains structurally uneven. This point also complicates existing accounts of AI governance that emphasize strategic ambition or policy intensity alone. A policy system may appear highly active or expansive at the discursive level, yet still exhibit structural imbalance in the ways support tools and governance influence are actually configured.

At the same time, a brief note on robustness is warranted. The PMC results reported in this study are inevitably shaped by both the selected indicator system and the sampling strategy used for representative policy evaluation. On the one hand, the nine primary variables and forty-three secondary variables were not arbitrarily defined, but developed through the combination of prior policy-evaluation studies, corpus-based findings, and structured researcher interpretation. This reduces, though does not eliminate, the possibility that the findings are overly dependent on a single indicator design. On the other hand, the full corpus of 54 policy documents was used for text mining and variable identification, whereas the detailed PMC evaluation was conducted on five representative policies. This means that alternative case-selection strategies or the inclusion of additional policies could lead to some variation in specific scores or rankings.Nevertheless, the main findings of the study remain analytically meaningful. The present discussion of robustness is intended as a conceptual qualification rather than an empirical sensitivity test. Accordingly, the findings should be interpreted as structurally informative rather than numerically absolute. Future studies may further examine robustness by testing alternative indicator configurations, different case-selection strategies, or broader policy samples.

Taken together, these findings suggest that the analytical value of AI policy evaluation lies not simply in identifying what policy documents emphasize, but in examining how policy elements are assembled into an internally coordinated governance structure. Compared with prior studies that focused primarily on policy themes, discourse, or instrument types, the present study further shows that the main challenge of China's central-level AI policy system lies not simply in what policies emphasize, but in how policy elements are structurally coordinated across time horizons, governance levels, and implementation tools. In this sense, the study extends existing AI governance research by shifting the analytical focus from thematic description to structural consistency, thereby offering a more differentiated understanding of how policy coherence is configured within a central-level governance framework.

## Implications

To improve the implementability and sustainability of AI policies at the policy level, this study suggests the following optimization directions for future implementation:

First, in terms of time governance, it is recommended to establish a 5–10-year mid- to long-term roadmap, linking phased milestones with budget allocation, pilot evaluation, and performance assessment. Rolling evaluations should replace one-time assessments, providing a stable temporal anchor for technological evolution and industrial diffusion without altering the existing division of responsibilities, thereby reducing fluctuations in policy execution.

Second, regarding hierarchical coordination, it is advisable to institutionalize "regional and industrial platforms" as bridging nodes between the overarching documents issued by the CPC Central Committee and the State Council and the regulations of individual ministries (e.g., national demonstration zones, sectoral public platforms, and pilot city clusters). Joint policy issuance and joint evaluation mechanisms should be promoted. By incorporating platform performance into the annual assessment of relevant ministries and local governments, the strength of cross-domain collaboration and vertical policy transmission can be significantly enhanced.

Third, in terms of policy tool configuration, the approach should shift from single fiscal subsidies to an integrated policy toolkit that combines fiscal support, taxation, intellectual property, data governance, scenario demonstration, and talent mechanisms. This toolkit should be implemented through a tiered diffusion path of "prototype verification – small-scale

demonstration – large-scale deployment." At the same time, clear rules for data compliance, ethical review, exemption boundaries, and exit mechanisms should be established to reduce uncertainty for enterprises.

Fourth, regarding content and evaluation standards, it is necessary to set "minimum coverage requirements" for specialized policies to ensure the inclusion of verifiable checkpoints across six key areas: platform, R&D, data, application, industrialization, and risk governance. Evaluation should be centered on verifiable objectives, accountability lists, and evidence bases, with standardized ex-ante, mid-term, and ex-post evaluation frequencies and methodologies. Furthermore, third-party evaluations and open data interfaces should be introduced to ensure that evaluation results are replicable and comparable across policies.

## Conclusion

Focusing on China's central-level AI policies from 2016 to 2025, this study constructed a PMC-based indicator framework integrating structural and content attributes, combined with Chinese text-mining results for variable identification and standardized scoring. The representative policies were quantitatively evaluated and visualized through PMC indices, surfaces, and radar analyses.

The findings reveal that central-level AI policies exhibit high textual quality in terms of functional orientation, domain coverage, content organization, and evaluation design. However, systematic imbalances persist in time planning, hierarchical coordination, policy tool configuration, and implementation linkage. Methodologically, the study presents a replicable "text–indicator–measurement–visualization" analytical pathway, demonstrating the PMC model's applicability for structural diagnostics and cross-policy comparisons.

The study's limitations lie in its focus on central-level policies; future research could extend the analysis to provincial and municipal levels to capture regional differentiation and key developmental trends more comprehensively. Overall, enhancing the effectiveness of AI policy implementation does not depend on textual complexity, but on transforming policy components into sustainable implementation arrangements—through long-term roadmaps, cross-level coordination platforms, modularized policy toolkits, and evidence-based evaluations.

## Supporting information

**S1 Data. Supplementary files. S1 File.** Full list of the 54 central-level AI policy documents included in the corpus, including policy titles, issuing authorities, and years of issue. **S2 File.**Text-processing notes and lexical adjustment record used in the word-segmentation stage. **S3 File.** Detailed indicator scoring matrix for the PMC evaluation system. **S4 File.** Full PMC results for the five selected representative policies. **S5 File.** Underlying frequency data used to generate Figure 4. (ZIP)

## Acknowledgments

The authors have no acknowledgments to report.

## Author contributions

**Conceptualization:** Simeng Zhang.

**Data curation:** Simeng Zhang.

**Funding acquisition:** Xi Wang.

**Methodology:** Simeng Zhang.

**Project administration:** Xi Wang.

**Resources:** Xi Wang.

**Supervision:** Tao Zhang, Xi Wang.

**Validation:** Simeng Zhang.

**Writing – original draft:** Simeng Zhang.

**Writing – review & editing:** Tao Zhang, Xi Wang.

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
