## [Decision Letter · Decision Letter 0]

8 Feb 2026

PONE-D-25-58681Structural Consistency in AI Governance: A PMC Index Assessment with Evidence from China’s Central-Level PoliciesPLOS One

Dear Dr. zhang,

Thank you for submitting your manuscript to PLOS ONE. After careful consideration, we feel that it has merit but does not fully meet PLOS ONE’s publication criteria as it currently stands. Therefore, we invite you to submit a revised version of the manuscript that addresses the points raised during the review process. 

We look forward to receiving your revised manuscript.

Kind regards,

Farshid Danesh, Ph.D.

Academic Editor

PLOS One

Journal Requirements:

3. Please note that PLOS One has specific guidelines on code sharing for submissions in which author-generated code underpins the findings in the manuscript. In these cases, we expect all author-generated code to be made available without restrictions upon publication of the work. Please review our guidelines at https://journals.plos.org/plosone/s/materials-and-software-sharing#loc-sharing-code and ensure that your code is shared in a way that follows best practice and facilitates reproducibility and reuse.

4. Please provide additional details regarding participant consent. In the ethics statement in the Methods and online submission information, please ensure that you have specified (1) whether consent was informed and (2) what type you obtained (for instance, written or verbal, and if verbal, how it was documented and witnessed). If your study included minors, state whether you obtained consent from parents or guardians. If the need for consent was waived by the ethics committee, please include this information.

5. We note that your Data Availability Statement is currently as follows: All relevant data are within the manuscript and its Supporting Information files.

6. Please amend the manuscript submission data (via Edit Submission) to include author Xi Wang.

7. Please update your submission to use the PLOS LaTeX template. The template and more information on our requirements for LaTeX submissions can be found at http://journals.plos.org/plosone/s/latex.

8. Your abstract cannot contain citations. Please only include citations in the body text of the manuscript, and ensure that they remain in ascending numerical order on first mention.

9. Please ensure that you refer to Figure 1 in your text as, if accepted, production will need this reference to link the reader to the figure.

Reviewers' comments:

Reviewer's Responses to Questions

**Comments to the Author**

1. Is the manuscript technically sound, and do the data support the conclusions?

Reviewer #1: Yes

Reviewer #2: Partly

2. Has the statistical analysis been performed appropriately and rigorously? 

Reviewer #1: Yes

Reviewer #2: N/A

3. Have the authors made all data underlying the findings in their manuscript fully available?

Reviewer #1: Yes

Reviewer #2: No

4. Is the manuscript presented in an intelligible fashion and written in standard English?

Reviewer #1: Yes

Reviewer #2: Yes

5. Review Comments to the Author

Reviewer #1: This manuscript presents a timely and methodologically innovative evaluation of structural consistency in AI governance, making a valuable contribution to policy informatics and AI policy studies. The application of the PMC index model to China’s central-level AI policies is well motivated and executed, and the integration of text mining with structured policy evaluation represents a clear advancement over purely descriptive or thematic policy analyses.

The main strengths of the paper lie in its systematic indicator construction, transparent modeling logic, and policy-relevant findings—particularly the identification of deficiencies in temporal planning, vertical coordination, and incentive mechanisms. These insights are useful not only for China’s AI governance but also for broader comparative policy research.

To further strengthen the manuscript, I recommend the following:

1.Improve transparency and reproducibility by providing supplementary materials that document the policy corpus, keyword dictionaries, indicator scoring matrices, and full PMC results.

2.Clarify the interpretive framework of PMC scores, including what constitutes relatively strong or weak structural consistency in practical terms.

3.Briefly discuss robustness and potential sensitivity to indicator selection or policy sampling.

I have no concerns regarding dual publication, research ethics, or publication ethics. The study relies exclusively on public documents and poses no ethical risks.

Reviewer #2: 1- In the Introduction, the authors need to engage more deeply with the concept of artificial intelligence governance, clearly defining and theoretically framing this concept.

2- The Introduction should not include methodological details. All content related to the research methods should be removed from this section.

3- There is no need to describe the structure or sections of the paper in the Introduction, as this unnecessarily increases the length of the manuscript.

4- The Literature Review section is overly extensive and should be substantially condensed. It should focus primarily on the core topic of AI governance and the application of the PMC index. The concluding synthesis of this section should also be more concise and focused.

5- The data sources must be reported clearly and transparently, specifying exactly which databases or websites were used to collect the data.

6- The methodology is not described with sufficient clarity. It is unclear which parts of the texts were used as input for text mining, what types of text-mining analyses were conducted, the sequential steps of the analysis, and which tools and techniques were employed.

7- To improve transparency and rigor in reporting the text-mining methodology, the authors are encouraged to refer to the following study as a methodological guide:

Danesh F, Dastani M, Ghorbani M. Retrospective and prospective approaches of coronavirus publications in the last half-century: a Latent Dirichlet Allocation analysis. Library Hi Tech. 2021;39(3):855–872.

8- It is unclear whether the study relies solely on automated text-mining analyses or whether domain experts were involved in interpreting the results. For example, the analytical basis for Tables 2, 3, and 4 is not explained. This process must be described explicitly in the Methods section.

9- The Discussion section requires substantial rewriting. It should focus on the principal findings of the study and critically compare them with results from relevant previous research, rather than providing descriptive or repetitive content.

6. PLOS authors have the option to publish the peer review history of their article (what does this mean?). If published, this will include your full peer review and any attached files.

Reviewer #1: No

Reviewer #2: No

---

## [Author Response · Author response to Decision Letter 1]

2 Apr 2026

Response to Reviewers #1

（PONE-D-25-58681：Structural Consistency in AI Governance: A PMC Index Assessment with Evidence from China’s Central-Level Policies）

Thank you for your valuable comments and suggestions, which have greatly helped us improve our manuscript. We have carefully addressed each point and revised the paper accordingly. Below is our detailed response to your feedback:

1.“Improve transparency and reproducibility by providing supplementary materials that document the policy corpus, keyword dictionaries, indicator scoring matrices, and full PMC results. ”

Thank you very much for this constructive comment and for emphasizing the importance of transparency and reproducibility. In the revised manuscript, we have addressed this suggestion through both textual revision and supplementary documentation.

Specifically:

(1) In Section 3.3 Data Sources, we clarified the full policy corpus by specifying the 54 central-level AI policy documents included in the study, together with the retrieval sources and search strategy, including the official databases used for policy collection.

(2) In Section 3.4.1 Policy Text Mining and Word-Frequency Analysis, we further clarified the text-processing procedure in ROSTCM6.0, including corpus preparation, Chinese word segmentation, and the filtering of structurally repetitive or analytically uninformative expressions.

(3) In Section 3.4.2 Variable Classification and Parameter Identification, we expanded the explanation of the indicator scoring basis and supplemented a more detailed scoring matrix for the PMC evaluation system.

(4) We added a new subsection, Section “3.5 Transparency and Reproducibility Materials,” to explicitly describe the supplementary materials and their role in supporting the transparency, traceability, and reproducibility of the study.

(5) We also added a Data Availability Statement, indicating that the policy corpus list, text-processing notes (including the lexical resource used during segmentation and the filtering rules applied in preprocessing), the detailed indicator scoring matrix, and the full PMC results for the five selected representative policies are provided as supplementary materials.

In addition, we revised the related methodological descriptions to more accurately reflect the actual analytical workflow. In particular, rather than referring to a separately constructed standalone keyword dictionary, we now clarify that the segmentation process relied on the lexical resource available in the ROSTCM6.0 environment, followed by corpus-oriented filtering and manual checking. We believe these revisions substantially improve the transparency and reproducibility of the study.

We sincerely appreciate this constructive feedback, as it has helped us improve the transparency and reproducibility of the study by making the supporting materials more explicit and accessible.

2.“Clarify the interpretive framework of PMC scores, including what constitutes relatively strong or weak structural consistency in practical terms.”

Thank you for this insightful comment. We agree that PMC scores should not be presented only as numerical classifications, but should also be interpreted in substantive policy-design terms. In response, we revised the manuscript in the following two places:

(1)In Section 3.4.3 Calculation of the PMC Index, we added an interpretive paragraph after the PMC classification table to clarify that PMC scores reflect the structural consistency of policy design across the selected dimensions. We now explain that higher scores indicate a relatively more complete, balanced, and internally coordinated policy structure, whereas lower scores indicate a relatively less comprehensive or less balanced configuration.

(2)In Section 4.3.1 Overall Quality Assessment of China’s AI Policies, we further clarified the practical meaning of PMC score differences by linking them to the five representative policy cases. We explain that higher-scoring policies tend to show stronger structural alignment across major policy dimensions, whereas lower-scoring policies reveal relatively weaker balance or support in one or more dimensions.

We sincerely appreciate this constructive feedback, as it has helped us clarify the substantive meaning of PMC scores and strengthen the connection between numerical results and the practical interpretation of policy-structure consistency.

3.“Briefly discuss robustness and potential sensitivity to indicator selection or policy sampling.”

Thank you for this valuable comment. We agree that the robustness of the findings and their potential sensitivity to indicator selection and policy sampling should be discussed more explicitly. In response, we revised the manuscript in the following two places:

(1) In Section 5.1 Discussion, we added a paragraph discussing the potential sensitivity of the PMC results to both indicator selection and policy sampling. In the revised text, we clarify that the current indicator system was developed through the combination of prior policy-evaluation studies, corpus-based findings, and structured researcher interpretation, rather than arbitrary selection. We also explicitly note that the full corpus of 54 policy documents was used for text mining and variable identification, whereas the detailed PMC evaluation was conducted on five representative policies.

(2) At the end of this newly added discussion paragraph in Section 5.1, we included a limitation-oriented summary stating that the findings should be interpreted as structurally informative rather than numerically absolute, and that future research may further examine robustness by testing alternative indicator systems or broader policy samples.

Response to Reviewers #2

（PONE-D-25-58681：Structural Consistency in AI Governance: A PMC Index Assessment with Evidence from China’s Central-Level Policies）

Thank you for your valuable comments and suggestions, which have greatly helped us improve our manuscript. We have carefully addressed each point and revised the paper accordingly. Below is our detailed response to your feedback:

1.“In the Introduction, the authors need to engage more deeply with the concept of artificial intelligence governance, clearly defining and theoretically framing this concept.”

2.“The Introduction should not include methodological details. All content related to the research methods should be removed from this section.”

3.“There is no need to describe the structure or sections of the paper in the Introduction, as this unnecessarily increases the length of the manuscript.”

Thank you for these important comments. We agree that the Introduction should more clearly foreground the conceptual framing of artificial intelligence governance, while avoiding excessive methodological detail and unnecessary description of the paper structure. In response, we substantially revised the Introduction in the following two respects:

(1) We strengthened the conceptual and theoretical framing of the study by adding a clearer definition of artificial intelligence governance. In the revised Introduction, artificial intelligence governance is defined as the institutional, strategic, and regulatory arrangements through which the state guides, supports, coordinates, and constrains the development and application of AI. We also clarified why the structural consistency of policy design can serve as an analytical entry point for examining AI governance.

(2)We streamlined the Introduction by removing detailed methodological descriptions and the section-by-section outline of the paper. As a result, the revised Introduction now focuses more directly on the research background, the governance-related problem setting, the theoretical relevance of AI governance, and the overall research objective.

4.“The Literature Review section is overly extensive and should be substantially condensed. It should focus primarily on the core topic of AI governance and the application of the PMC index. The concluding synthesis of this section should also be more concise and focused.”

Thank you for this helpful comment. We agree that the original Literature Review was overly extensive and that it should be more tightly focused on the core themes of artificial intelligence governance and the PMC Index. In response, we revised this section in the following two respects:

(1)We substantially condensed the review of AI policy studies and reorganized it around the core theme of artificial intelligence governance. In the revised manuscript, this section now focuses more directly on the evolution of AI governance research, the main approaches used in existing AI policy studies, and the key limitation of prior work—namely, its emphasis on policy themes, discourse, and instruments rather than on the structural consistency of policy systems.

(2) We also condensed the review of the PMC Index Model by emphasizing its methodological relevance to structured policy evaluation, rather than providing an overly detailed listing of applications across many policy fields. In addition, we substantially shortened and sharpened the concluding synthesis of the Literature Review so that it now focuses more clearly on the specific research gap addressed by this study.

5.“The data sources must be reported clearly and transparently, specifying exactly which databases or websites were used to collect the data.”

Thank you for this helpful comment. We agree that the data sources should be reported more explicitly and transparently. In response, we revised the manuscript in the following two respects:

(1) In Section 3.3 Data Sources, we clarified the exact sources used to collect the policy documents. The revised manuscript now explicitly specifies that the corpus was retrieved primarily from the Chinese Government Website (www.gov.cn) and PKUlaw (pkulaw.com), and also reports the data retrieval period and the main screening principles applied during corpus construction.

(2) To further enhance transparency, we provide the full list of the 54 central-level AI policy documents as S1 File, so that readers can directly verify the composition of the policy corpus used in this study.

We sincerely appreciate this constructive feedback, as it has helped us improve the transparency and traceability of the study by making the policy data sources and corpus construction process more explicit.

6.“The methodology is not described with sufficient clarity. It is unclear which parts of the texts were used as input for text mining, what types of text-mining analyses were conducted, the sequential steps of the analysis, and which tools and techniques were employed.”

7.“To improve transparency and rigor in reporting the text-mining methodology, the authors are encouraged to refer to the following study as a methodological guide:Danesh F, Dastani M, Ghorbani M. Retrospective and prospective approaches of coronavirus publications in the last half-century: a Latent Dirichlet Allocation analysis. Library Hi Tech. 2021;39(3):855–872.”

Thank you for these important comments. We agree that the methodological workflow, especially the text-mining component and its relation to the PMC coding process, should be described more explicitly. In response, we revised the Methods section in the following three respects.

(1) We substantially revised Sections 3.2 and 3.4.1 to clarify the methodological sequence of the study. The revised manuscript now explicitly states that the full texts of the 54 policy documents were used as the input for text mining, and that the text-mining stage consisted of corpus preparation, Chinese word segmentation, word-frequency analysis, and semantic co-occurrence analysis using ROSTCM6.0 and NetDraw. We also clarified that these procedures were used to identify thematic features of the corpus and support variable construction, rather than to generate final PMC scores automatically.

(2) We revised Section 3.4.2 to explain more clearly how Tables 2, 3, and 4 were constructed. In the revised text, we specify that the indicator system in Table 2 was developed through the combination of prior policy-evaluation studies, corpus-based text analysis, and researcher interpretation of China’s AI policy texts. Table 3 is now described as the operationalized multi-input–output structure derived from Table 2, and Table 4 is explicitly presented as the PMC classification standard adopted from prior research.

(3) We further clarified that the study does not rely solely on automated text-mining analysis. Rather, it combines automated corpus processing with researcher-led interpretation and binary coding based on a predefined scoring matrix. To improve transparency in reporting the text-mining workflow, we also referred to the methodological reporting logic of Danesh et al. (2021) as a guide when revising this part of the manuscript, particularly in specifying the input corpus, preprocessing procedures, analytical sequence, and the relationship between automated analysis and researcher-led interpretation. In addition, the relevant supporting materials are now provided in the supplementary files.We sincerely appreciate this constructive feedback, as it has helped us strengthen the methodological transparency, interpretive clarity, and reproducibility of the study.

8.“It is unclear whether the study relies solely on automated text-mining analyses or whether domain experts were involved in interpreting the results. For example, the analytical basis for Tables 2, 3, and 4 is not explained. This process must be described explicitly in the Methods section.”

Thank you for this important comment. We agree that the original manuscript did not explain with sufficient clarity the respective roles of automated text mining and researcher interpretation, nor did it make the analytical basis of Tables 2, 3, and 4 sufficiently explicit. In response, we revised the Methods section in a more systematic way, so that the analytical workflow and the basis of each key table are now clearly specified.

(1) In Section 3.2 Research Method, we clarified that the study does not rely solely on automated text-mining analysis. Instead, the methodological process is sequential: text mining was first used to explore the thematic and semantic features of the full policy corpus, and the results were then integrated with prior policy-evaluation research, corpus-based interpretation, and structured researcher judgment to construct the PMC analytical framework. We now explicitly state that text mining supported corpus exploration and variable identification, but did not automatically generate the final indicator system or PMC scores.

(2) In Section 3.4.1 Policy Text Mining and Word Frequency Analysis, we further specified that the input for text mining consisted of the full texts of the 54 central-level AI policy documents, rather than only titles or selected excerpts. We also clarified the analytical sequence of the text-mining stage, including corpus preparation, Chinese word segmentation, word-frequency analysis, and semantic co-occurrence analysis using ROSTCM6.0 and NetDraw. This revision makes the technical workflow of the text-mining component more transparent.

(3) In Section 3.4.2 Variable Classification and Parameter Identification, we explicitly explained that Table 2 is not a direct output of automated text mining. Rather, it is a researcher-constructed indicator system developed through the combination of corpus-based findings, prior policy-evaluation literature, and structured interpretation of China’s AI policy texts. We also clarified that the detailed coding basis for all secondary indicators is provided in the supplementary scoring matrix.

(4) In Section 3.4.3 Construction of the Multi-Input–Output Table, we clarified that Table 3 operationalizes the indicator system presented in Table 2 by reorganizing the primary and secondary variables into a standardized binary coding structure for PMC evaluation. This makes clear how the indicator system was translated into the coding framework used for subsequent calculation.

(5) In Section 3.4.4 Calculation of the PMC Index, we explained that Table 4 serves as the interpretive classification standard for PMC scores, adopted from prior studies, rather than as a direct output of text mining or automatic coding. In o

---

## [Decision Letter · Decision Letter 1]

4 May 2026

PONE-D-25-58681R1Structural Consistency in AI Governance: A PMC Index Assessment with Evidence from China’s Central-Level PoliciesPLOS One

Dear Dr. wang,

Thank you for submitting your manuscript to PLOS ONE. After careful consideration, we feel that it has merit but does not fully meet PLOS ONE’s publication criteria as it currently stands. Therefore, we invite you to submit a revised version of the manuscript that addresses the points raised during the review process.

We look forward to receiving your revised manuscript.

Kind regards,

Farshid Danesh, Ph.D.

Academic Editor

PLOS One

Journal Requirements:

Reviewers' comments:

Reviewer's Responses to Questions

**Comments to the Author**

1. If the authors have adequately addressed your comments raised in a previous round of review and you feel that this manuscript is now acceptable for publication, you may indicate that here to bypass the “Comments to the Author” section, enter your conflict of interest statement in the “Confidential to Editor” section, and submit your "Accept" recommendation.

Reviewer #1: All comments have been addressed

Reviewer #3: (No Response)

2. Is the manuscript technically sound, and do the data support the conclusions?

Reviewer #1: Yes

Reviewer #3: Partly

3. Has the statistical analysis been performed appropriately and rigorously? 

Reviewer #1: Yes

Reviewer #3: Yes

4. Have the authors made all data underlying the findings in their manuscript fully available?

Reviewer #1: Yes

Reviewer #3: Yes

5. Is the manuscript presented in an intelligible fashion and written in standard English?

Reviewer #1: Yes

Reviewer #3: (No Response)

6. Review Comments to the Author

Reviewer #1: The responses are complete, precise, and fully address the reviewer’s concerns.

Each point is answered with clear revisions, explicit section references, and concrete modifications to the manuscript.

The replies demonstrate transparency, methodological clarity, and direct alignment with the reviewer’s requests.

Overall, the response document is strong and should be satisfactory for the re‑review

Reviewer #3: The revised manuscript has improved substantially compared with the previous version. The authors have addressed most of the previous reviewers’ comments satisfactorily. In particular, the introduction now provides a clearer conceptual framing of AI governance, the literature review is more focused, the data sources are more transparent, the methodology is better explained, and the interpretation of PMC scores has been strengthened.

The manuscript is generally technically sound, and the data largely support the main conclusions. The authors have clarified that the full texts of the 54 policy documents were used as input for text mining and that the analytical process included corpus preparation, Chinese word segmentation, word-frequency analysis, and semantic co-occurrence analysis using ROSTCM6.0 and NetDraw. They have also clarified that text mining supported variable construction rather than automatically generating the final PMC scores. This substantially improves the methodological transparency of the study.

The authors have also improved the explanation of the PMC scoring framework. The revised manuscript now explains that higher PMC scores indicate a more complete, balanced, and internally coordinated policy structure, whereas lower scores indicate weaker coverage or imbalance across policy dimensions. This makes the interpretation of the quantitative results clearer and more meaningful in policy terms.

However, a few minor issues still require attention. First, the completeness and adequacy of the supplementary materials should be carefully verified. The revised manuscript refers to supplementary files documenting the full policy corpus, text-processing notes, scoring matrix, full PMC results, and underlying figure data. These materials are important for transparency and reproducibility, and they should be complete and clearly organized.

Second, the discussion of robustness and sensitivity remains mostly conceptual. The authors acknowledge that the results may be affected by the selected indicator system and the choice of five representative policies, and this clarification is useful. However, no empirical robustness or sensitivity analysis has been conducted. If possible, the authors should either add a limited sensitivity check or more clearly state this as a limitation.

Third, the authors state in their response that they used the methodological reporting logic of Danesh et al. (2021) to revise the text-mining section. However, a clear full citation to this study was not identified in the revised manuscript or reference list. If this study was used as a methodological guide, the full citation should be added to the manuscript and the reference list.

Fourth, the Discussion section has improved and is now better organized around the main findings of the study. Nevertheless, the critical comparison with previous studies could still be slightly strengthened. The authors may add a few more explicit statements explaining how their findings confirm, refine, or differ from prior research on AI governance and PMC-based policy evaluation.

Overall, the revised manuscript responds well to the previous reviewers’ comments. The main concerns have been addressed, and only minor revisions remain necessary.

7. PLOS authors have the option to publish the peer review history of their article (what does this mean?). If published, this will include your full peer review and any attached files.

Reviewer #1: No

Reviewer #3: No

---

## [Author Response · Author response to Decision Letter 2]

10 May 2026

Response to Previous Reviewer #1

Comment 1

Evaluation Outcome

“This concern has been largely addressed. The revised manuscript provides clearer explanations of where the data were obtained, how the policy texts were processed, and how the indicators were operationalized. The addition of a transparency and reproducibility section strengthens the methodological reporting of the study. However, the completeness and adequacy of the supplementary materials should still be verified to ensure that they actually include the full policy corpus, the detailed scoring matrix, and the complete PMC results.”

Thank you for this helpful evaluation. We agree that, beyond the revisions made in the main manuscript, the completeness and adequacy of the supplementary materials should also be explicitly verified. In response, we re-checked all supporting files and ensured that they correspond directly to the main stages of the analytical workflow. Specifically, S1 File provides the full list of the 54 central-level AI policy documents included in the corpus, including their issuance years; S2 File provides the text-processing notes and lexical adjustment record used in the word-segmentation stage; S3 File provides the detailed indicator scoring matrix for the PMC evaluation system; S4 File provides the full PMC results for the five selected representative policies; and S5 File provides the underlying frequency data used to generate Figure 4. In addition, the binary coding results, primary-variable scores, PMC indices, and PMC matrices for the five representative policies are reported in Tables 6–8 of the main manuscript. We also clarified in the revised Data Availability Statement that the yearly distribution shown in Figure 3 is based on the corpus information provided in S1 File. These revisions were intended to ensure that the supplementary materials fully support the transparency, traceability, and reproducibility of the study.

Comment 3

Evaluation Outcome

“This concern has been partially addressed. The authors have discussed robustness and sensitivity at a conceptual level, which is acceptable given that the previous comment requested a brief discussion. However, the manuscript still does not include an empirical robustness test or sensitivity analysis. Therefore, this issue has been addressed at the discussion level, but not fully resolved empirically.”

Thank you for this helpful evaluation. We agree that the current manuscript addresses robustness and potential sensitivity mainly at the conceptual discussion level rather than through an empirical robustness test. In light of the original review request, which asked for a brief discussion of robustness and sensitivity, we retained this discussion-oriented treatment but further clarified its scope in the revised manuscript. Specifically, we now state more explicitly that the robustness discussion is intended as a conceptual qualification rather than an empirical sensitivity analysis, and that the findings should therefore be interpreted as structurally informative rather than numerically absolute. We also emphasize that future research may examine robustness more systematically by testing alternative indicator configurations, different case-selection strategies, or broader policy samples.

Response to Previous Reviewer #2

Comment 7

Evaluation Outcome

“This concern has been partially addressed. The reporting of the text-mining procedure has improved. However, a clear full citation to Danesh et al. (2021) was not identified in the revised manuscript or reference list. If the authors used this study as a methodological reporting guide, the full citation should be added to the text and reference list.”

Thank you for this helpful evaluation. We agree that, if Danesh et al. (2021) is referred to as a methodological reporting guide in our previous response, the corresponding citation should also be clearly included in the revised manuscript and reference list. In response, we have now added an explicit citation to Danesh et al. (2021) in Section 3.2 Research Method to indicate that the reporting of the text-mining procedure was further refined with reference to prior methodological studies on transparent text-analytic workflows. We have also added the full bibliographic information to the reference list. This revision was intended to make the methodological basis of our reporting more explicit and consistent between the manuscript and the response letter.

Comment 9

Evaluation Outcome

“This concern has been largely addressed. The revised Discussion section is more analytical and better organized around the main findings. It also links the findings more clearly to previous research. However, the critical comparison with prior studies could still be slightly strengthened.”

Thank you for this helpful evaluation. We agree that the revised Discussion section could be further improved by making the comparison with prior studies more explicit. In response, we have slightly strengthened the concluding part of Section 5.1 Discussion. Specifically, the revised text now states more directly that, compared with prior studies focusing mainly on policy themes, discourse, or instrument types, the present study further highlights the structural coordination problem of China’s central-level AI policy system across time horizons, governance levels, and implementation tools. We also clarify that the study extends existing AI governance research by shifting the analytical focus from thematic description to structural consistency. These revisions were intended to make the comparative and critical contribution of the Discussion more explicit.

---

## [Editor Report · Decision Letter 2]

11 May 2026

Structural Consistency in AI Governance: A PMC Index Assessment with Evidence from China’s Central-Level Policies

PONE-D-25-58681R2

Dear Dr. wang,

We’re pleased to inform you that your manuscript has been judged scientifically suitable for publication and will be formally accepted for publication once it meets all outstanding technical requirements.

Kind regards,

Farshid Danesh, Ph.D.

Academic Editor

PLOS One
---

## [Editor Report · Acceptance letter]

PONE-D-25-58681R2

PLOS One

Dear Dr. wang,

I'm pleased to inform you that your manuscript has been deemed suitable for publication in PLOS One. Congratulations! Your manuscript is now being handed over to our production team.

Kind regards,

on behalf of

Associate Professor Farshid Danesh

Academic Editor

PLOS One